# Two-edge-resolved three-dimensional non-line-of-sight imaging with an ordinary camera

Robinson Czajkowski[1] & John Murray-Bruce [1] ✉

We introduce an approach for three-dimensional full-colour non-line-of-sight imaging with an ordinary camera that relies on a complementary combination of a new measurement acquisition strategy, scene representation model, and tailored reconstruction method. From an ordinary photograph of a matte line-of-sight surface illuminated by the hidden scene, our approach reconstructs a three-dimensional image of the scene hidden behind an occluding structure by exploiting two orthogonal edges of the structure for transverse resolution along azimuth and elevation angles and an information orthogonal scene representation for accurate range resolution. Prior demonstrations beyond two-dimensional reconstructions used expensive, specialized optical systems to gather information about the hidden scene. Here, we achieve accurate three-dimensional imaging using inexpensive, and ubiquitous hardware, without requiring a calibration image. Thus, our system may find use in indoor situations like reconnaissance and search-and-rescue.

Conventional imaging and vision systems require a direct line of sight of the scene of interest. However, in numerous applications like search-and-rescue, reconnaissance, infrastructure evaluation, archaeological expeditions, and biomedical imaging, obtaining a direct line of sight may be unsafe, challenging, or even impossible. This has proliferated in a number of research fields aiming to image through or around obstacles, by leveraging different sensing modalities including optical[1–19], sound[20,21], thermal[22] and others[23,24]. The rapidly growing field of non-line-of-sight (NLOS) imaging[1] is one such field that aims to resolve this challenge and extend the limits of conventional optical systems, by decoding information in measurements of the light that reaches the visible side after being reflected or emitted by the hidden scene. With the ability to see around obstacles, these systems will profoundly impact myriad applications across different fields.

Conceptualised by ref. 2, the first experimental demonstration of 3D NLOS imaging used active optical illumination of a visible surface and time-resolved detection of the returning light after suffering three diffuse reflection events[3]: from the visible surface, hidden scene surfaces and visible surface again, in that order. These and subsequent active 3D NLOS imaging works, raster scan the illumination[4,25], or the time-resolved detector[26,27], or both in a confocal setup[6–9] over a large 2D grid of points to gather measurements containing sufficient information about the 3D structure of the hidden scene. Most active NLOS methods are thus, inherently limited by slow acquisition speeds due to the requirement of raster scanning. Excluding the recent demonstration[28], active methods that use a detector array instead of raster scanning are constrained to tracking one or two moving hidden scene components[29,30]. Impressively, ref. 28 tracks one or two moving hidden scene components as rectangular facets and simultaneously maps out the static background behind them from comparatively short acquisition times, although the requirement of a long-exposure calibration measurement and inability to resolve shape, pose, or full 3D information may limit its applicability. Moreover, active methods that use intensity-only information are similarly limited to detecting and roughly tracking a single NLOS object[18].

Contrasting their active counterpart, passive NLOS imaging methods do not use controlled illumination, opting instead to use existing light arriving naturally from the hidden scene for reconstruction. Here, the detected light suffers at least one diffuse reflection event and attenuation according to an inverse-square law, which destroys

[1]Department of Computer Science and Engineering, University of South Florida, 4202 E. Fowler Avenue, Tampa, FL 33620, USA.
✉e-mail: murraybruce@usf.edu

most information about light path directions and produces a weak signal. The most promising passive methods that use ordinary cameras as detectors aim to restore directional information by exploiting the edges of an occluding structure existing in both the hidden and visible scenes[13,31–35], or entirely in the hidden scene[14–16] and are to be estimated or assumed to be known. In a typical scenario, the illumination of the visible surface caused by light from the hidden scene depends on its interaction with the objects preventing a direct line of sight. Some light paths are obstructed by the occluding objects, whereas other light paths are unobstructed and, thus, illuminate the visible scene area. This interaction produces, on the surfaces of the visible measurement area, a soft shadow (or penumbra)[36] that is informative about both the occluding structure and the scene it obscures[16,36].

Although significant strides have been made, notably by the class of works based on the corner camera[32] that exploit vertical edges— such as those occurring in corridors, doorways, or at the boundaries of buildings in urban environments—to form reconstructions of a hidden scene[33,34,37], intensity-based passive methods (including those that exploit hidden scene occluders[14–17,38]) are limited to 1D and 2D reconstructions. Bouman et al.[32] reconstructed 1D videos (azimuthal trajectories) of moving hidden scene components from a video recording of penumbra occurring on the floor, while a subsequent work[37] reconstructed both moving and stationary hidden scene components in 1D, from single photographs. These prior works[32,37] and their extensions[28,33,34,39,40] derive high azimuthal resolution from the vertical edge separating light path directions in azimuth. Adding a second dimension to obtain 2D plan view reconstructions relied either on measured travel times of ultrashort light pulses used to probe the hidden scene area[39,40], a stereo combination of corner cameras[32], or on detailed modelling and sophisticated scene priors[28,33], with varying degrees of accuracy.

In this work, we present a completely passive approach which we dub two-edge-resolved imaging (TERI), because of its opportunistic use of two orthogonal edges of a visible occluder structure to achieve 3D full-colour imaging of a hidden scene with an ordinary camera. Such two-edge instances in occluders are abundant: appearing at the top of doorways and window frames in indoor settings. The opportunistic use of such visible edge occluders by TERI is akin to the use of vertical edges by corner cameras[32,33,37,40]. In contrast, however, we propose a new acquisition configuration (see Fig. 1) that uses the visible ceiling as the observation plane. In our configuration, it is less likely that contributions from the most interesting hidden scene components, which typically rest on the ground plane, will be overwhelmed by bright overhead ambient light contributions. This is because direct contributions from elements of the hidden scene ceiling, which tend to be bright illumination sources, are completely occluded by the door frame head (see Supplementary Fig. 4). Moreover, observing the ceiling plane enables the vertical and horizontal edges of the occluding door (or window) frame structure to be simultaneously exploited for high azimuthal and elevation angle resolution, respectively. By combining sophisticated modelling, an explicit scene prior, and a theoretically grounded scene representation, we glean highly accurate range information from the measured photographs—effectively recovering the previously elusive third dimension. TERI's proposed scene representation also yields closed-form expressions that facilitate the efficient formation of light transport matrices and estimation of full-colour 3D imagery. Consequently, TERI achieves comparable visual reconstruction quality to state-of-the-art active 3D NLOS methods with the added benefit of a fast acquisition time.

## Results

### Passive NLOS imaging configuration

Edges are a common feature in many real-life settings: existing as the boundaries of ubiquitous structures like buildings, vehicles, furniture, doorways, window frames, and so on. Often, particularly in urban scenarios, the object occluding the direct line-of-sight to a desired

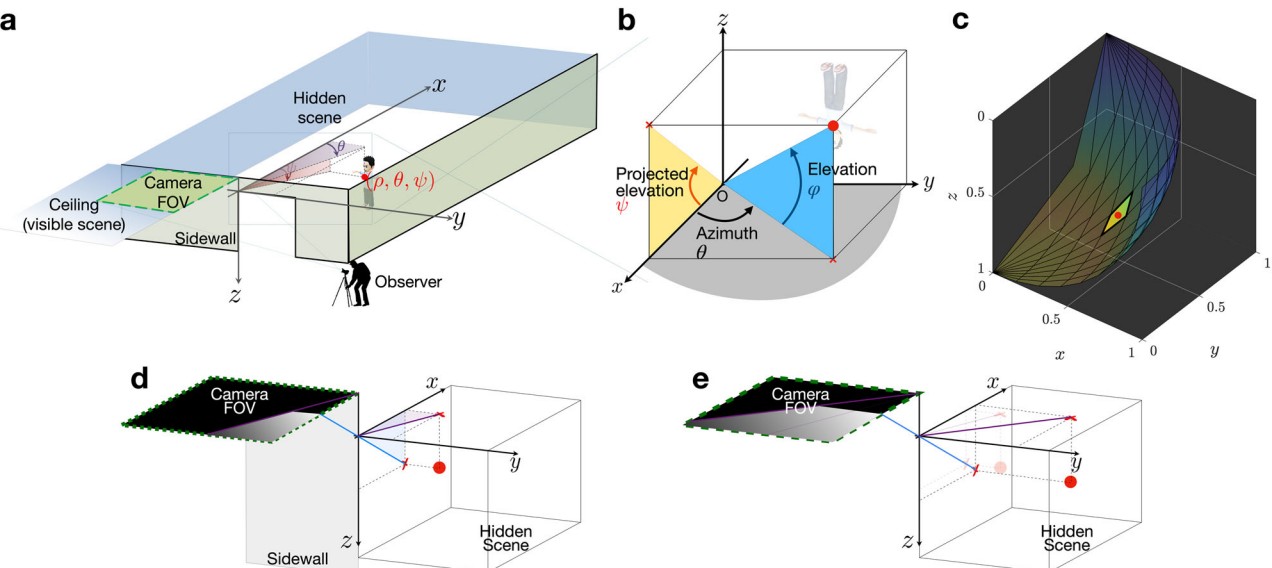

**Fig. 1 | Two-edge-resolved NLOS imaging scenario and hidden scene representation. a** Depiction of the imaging scenario and proposed projected-elevation spherical coordinate. With the origin at the upper-left corner of the door frame, a hidden scene point is identified by its range $\rho$, azimuth $\theta$, and projected-elevation $\psi$. **b** Shows the projected-elevation $\psi$ in the proposed projected-elevation spherical coordinate system, it is the projection of the conventional elevation angle of spherical coordinates onto the $xz$-plane and is such that $\tan(\psi) = \tan(\varphi)\sec(\theta)$. (For clarity, the $z$-axis is flipped from (**a**) to point upward.) **c** Elemental surface representation resulting from 10 equal divisions of azimuth and projected-elevation axes with fixed range, $\rho$. Indicated by the red dot is an example surface element whose centre is at $(\rho, \theta, \psi) = (1, 11\pi/40, 13\pi/40)$ and angular extents equal $\pi/20$ along azimuth and projected-elevation. **d, e** Depict the changes in the observed measurement due to a hidden point source (red dot) moving from its position in (**d**) to a new position in (**e**) such that its range and projected-elevation angle are fixed and only its azimuthal angle changes. The light from a hidden scene point is occluded by the doorway edges to create an illuminated region of trapezoidal shape on the ceiling. The observation in (**d**) has an illuminated trapezoidal region whose slanted edge is steeper than that of (**e**) because the azimuthal angle of the point source increases from (**d**) to (**e**); the heights of the illuminated trapezoid portions in (**d**) and (**e**) are otherwise equal because the projected-elevation angle is unchanged.

scene has edges that are visible both to the observer and the hidden scene. Figure 1a illustrates such a scenario and the proposed acquisition configuration of TERI for passive 3D NLOS reconstructions.

Imaging the ceiling with an ordinary camera, instead of the floor as in prior corner-camera-based demonstrations[32,33,37,40,41], allows both the vertical and horizontal edges of the occluding doorway to be exploited for NLOS imaging with high azimuthal and elevation angle resolution. The mechanism producing these two dimensions of high resolution is illustrated in Fig. 1d, e. Any hidden scene point in isolation creates a contribution, to the observation plane region enclosed by the green dashed line in Fig. 1d and e, which comprises an unobscured portion in the shape of a right trapezoid and an obscured portion (shadow) due to the occluding wall and doorway head. The sharp shadow makes the azimuthal and elevation angles of a hidden scene point easily recoverable, because the slope of the slanted shadow edge in the observation depends on the azimuthal angle, while the height of the right trapezoid depends on the elevation angle component of the hidden scene point. Hence, hidden scene points with unique azimuth and elevation angles will produce corresponding unique right trapezoidal illuminations of the observation plane. The geometry of the right trapezoid is dependent on the azimuth and elevation of the hidden scene point. Narrated animations provided in Supplementary Movie 1 further illustrate this dependence. This phenomenon enables high 2D angular resolution in estimating the hidden scene from the observed shadow. The utility of the second edge in providing high elevation angle resolution is also demonstrated by the Cramér-Rao Bound analyses provided in Supplementary Note 2.

Through simple geometric calculations, the azimuth and elevation angles of a hidden scene point can be estimated from the slope and height of the illuminated right trapezoidal region of the observation. However, spatially extended hidden scenes do not create sharp shadows. They instead produce a superposition thereof, that is dependent on the geometry of the scene from the vantage of the observed ceiling portion, appearing as a region of highly informative penumbra[36] (see Fig. 2a for an example photograph of penumbra). The penumbra region encodes information about the geometry of the hidden scene from the perspective of the ceiling, along two dimensions (i.e., in azimuth and elevation angles). Efficiently extracting this information for reconstructing a general spatially extended scene is achievable through a computational algorithm presented later on.

While the occluding edges of the doorway enable high angular resolution, no similar mechanisms exist to make the hidden scene point's radiosity and range easily recoverable with high resolution. They must both be estimated from the subtle intensity gradations present in the unobscured portion of the observation. To that end, we propose a specially crafted and theoretically motivated coordinate system and hidden scene representation, accompanied by a two-stage computational algorithm to enable reliable estimation of both

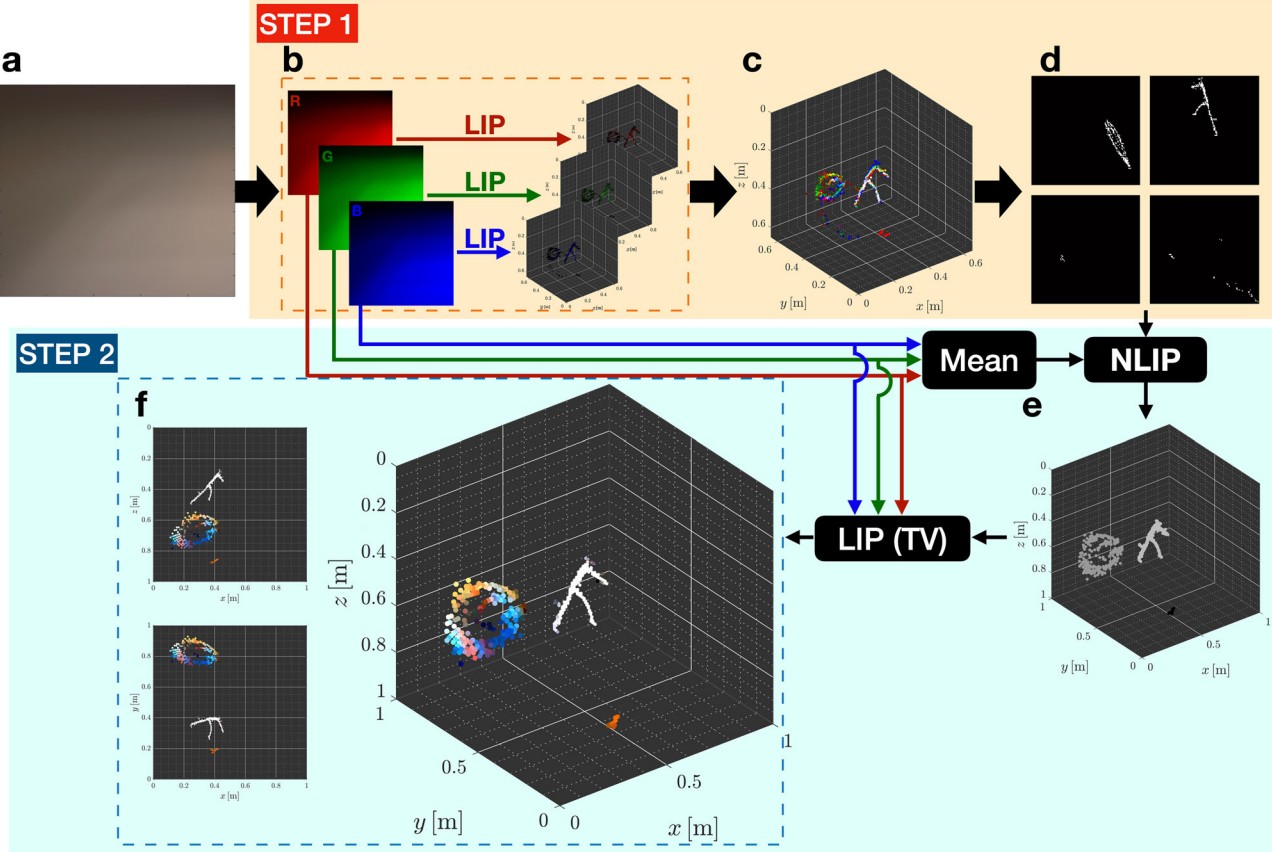

**Fig. 2 | Two-step reconstruction procedure. a** Measured penumbra photograph. **b** A linear inverse problem (LIP) is solved, per colour channel, to recover the azimuth and projected-elevation representation (i.e., the shapes) of hidden scene objects, with the entire hidden scene assumed to be confined to a single fixed range. **c** Colour visualisation of the initial angular reconstruction. **d** The reconstruction is analysed for connected surface elements which most likely belong to the same cluster. Four such clusters were identified, one for each of the three objects and a fourth (bottom right) for spurious elements. **e** A non-linear inverse problem (NLIP) is solved to estimate four ranges and four global radiosities, one for each cluster identified in (**d**). **f** Multiple views of the 3D colour reconstruction produced by incorporating the estimated ranges and solving a resulting total-variation-regularised LIP to obtain smoother estimates; a 3D view (second column), plan view (first column, top) and side view (first column, bottom) of the final reconstruction. (Best viewed in colour. The reconstruction procedure is also explained in Supplementary Movie 1).

radiosity and range, in addition to shape (i.e., azimuthal and elevation angle extents) of the hidden scene.

## Hidden scene representation

The hidden scene is assumed to be a collection of elemental surfaces, each with an unknown radiosity (brightness) and unique 3D position, to be estimated. This is the basis of the numerous analytical 3D scene visualisation techniques in computer vision and graphics. However, the desired representation is one that enables the most reliable reconstruction of the hidden scene from the measured NLOS photograph.

Here, our choice of the form and positions of the elemental surfaces is designed to make the unknown (positional) parameters of the hidden scene information orthogonal[42, p. 49] with respect to the Fisher information (FI) metric. The FI metric quantifies the informativeness of a set of measurements about a set of unknowns. Moreover, establishing Fisher information orthogonality among a set of unknown parameters guarantees asymptotic independence of their maximum likelihood estimates, which, crucially to the success of our approach, implies that the asymptotic variance of the estimate of a parameter does not depend on knowing any of the other parameters.

To achieve this property, we identify any point in the hidden scene by its range $\rho$, azimuthal angle $\theta$, and projected-elevation angle $\psi$ positions, with the origin at the apex of the vertical and horizontal edges of the doorway, as shown in Fig. 1b. The azimuth $\theta$ is the angle between the positive $x$-axis and the line segment connecting the origin to the $xy$-plane projection of the hidden scene point. Analogously, the projected-elevation $\psi$ is the angle between the positive $x$-axis and the line segment connecting the origin to the $xz$-plane projection of the hidden scene point. Because $\psi$ is the $xz$-plane projection of the conventional elevation angle used in spherical coordinates, this system is called projected-elevation spherical coordinates. Transforming to and from Cartesian coordinates is achievable via a set of expressions that are derived in Supplementary Note 1 (S1.2). Refer to Fig. 1b for a visualisation of the projected-elevation angle in this coordinate system and the conventional elevation angle of a spherical coordinate system.

Among all possible choices of coordinate systems, the proposed projected-elevation spherical coordinate system leads to a natural representation of the hidden scene where almost all parameters of the hidden scene will be information orthogonal. A geometric depiction of the information orthogonality property among the angular parameters, $\theta$ and $\psi$, is provided in Fig. 1d, e, as well as in Supplementary Movie 1. To achieve information orthogonality, the proposed coordinate system is constructed such that its angular coordinates, $(\theta, \psi)$, each pivot about one of the vertical and horizontal occluding edges of the doorway. Much like how polar (or cylindrical) coordinates are natural for the corner camera[32] and its variants[33,37,40], the projected-elevation spherical coordinate system is natural for TERI. Because single-edge corner cameras exploit a vertical occluding edge for resolution, a point source moving in the hidden scene changes only the observed penumbra shape when it moves only along the azimuth (pivoting about the vertical edge). Conversely, range-only motion produces subtle intensity changes in the penumbra, due to radial fall-off effects, while the overall penumbra shape is unchanged. The projected-elevation spherical coordinate system, therefore, represents a combination of two orthogonal corner camera systems resulting from two orthogonal edges of the doorframe. In this two-edge configuration, a hidden scene point source will produce a sharp shadow resembling the top and slanted edges of a right trapezoid (see example observations shown in Fig. 1d, e). Representing the hidden scene point in the proposed coordinate system, changes in the slope of the slanted edge of the shadow are due exclusively to point source motion along azimuth $\theta$ (with pivot at the vertical occluding edge). Conversely, changes in the position of the top edge of the shadow are due exclusively to source motion along the projected-elevation angle $\psi$ (with

pivot at the horizontal occluding edge). This apparent decoupling of how changes in $\theta$ and $\psi$ affect the measurement ensures that errors in any one of the angular coordinates will not impact our estimation of the other. Because the elevation is defined in the plane containing the azimuth, however, this phenomenon does not occur in a spherical coordinate system.

In addition, variations in the range or radiosity of the object have no impact on the obstructed non-illuminated portion of the observation; it only affects the brightness of the illuminated portion. An overall brightness change in the illuminated portion of the observation plane, with no change to the slope and height of the right trapezoidal region, is explainable by a corresponding change in the range and/or radiosity of the hidden scene point. This suggests that, although highly coupled themselves, the radiosity and range parameters have no coupling with the pair of angular parameters. Thus, errors in range or radiosity have a negligible impact on the angular estimates. No convenient (re-)parameterisations exist that make the range and radiosity parameters similarly information orthogonal. However, we alleviate the effect of their coupling through a two-step computation that first estimates the radiosities and scene shape (2D angular) components while assuming that the entire scene is confined to a fixed range, and subsequently computes ranges of the recovered hidden scene objects. Central to this approach is the assumption that neighbouring clusters of surface elements belong to the same object and thus have approximately equal ranges. This results in the significantly more feasible problem of recovering a single range for clusters of estimated surface elements, in place of the possibly intractable one that estimates a range per recovered surface element.

The existence of information orthogonality among the range, azimuth, and project-elevation coordinates is crucial to the success of our two-step reconstruction procedure. Notably, it ensures that accurate angle estimation (i.e., shape) is achievable without knowing the ranges of objects within the hidden scene because errors in one do not prevent reliable reconstruction of the other. Additionally, the proposed coordinate system also simplifies the forward modelling by enabling closed-form expressions for incorporating occlusion. Consequently, we could accurately compute the forward model matrix without performing computationally intensive numerical integrations. We formalise these claims and elucidate the information orthogonality phenomenon through Fisher information analyses, and experimental demonstrations in Supplementary Notes 2 and 3.

## Light transport model

The amount of light received by a point on the observation plane from a hidden scene point is well-modelled by a cumulative product of factors accounting for the radiosity of the hidden scene point, the visibility between the observed ceiling point and scene point, the fall-off in intensity due to the distance between both points and the effects of foreshortening and Lambertian reflection[14,43]. Adopting the proposed projected-elevation spherical coordinate hidden scene representation, and retaining a Cartesian representation of the measurement plane, the hidden scene point $\mathbf{s} = (\rho, \theta, \psi)$ with radiosity $c$, range $\rho$, azimuth $\theta$, and projected-elevation $\psi$, produces a contribution to any point $\mathbf{p} = (p_x, p_y, -h)$ on the visible ceiling plane $h$ metres above the doorway's edge, given by:

$$\ell(\mathbf{p}; \rho, \theta, \psi) = c \frac{\cos\left(\angle(\mathbf{p} - \mathbf{s}_c, \mathbf{n_p})\right)}{\|\mathbf{p} - \mathbf{s}_c\|_2^2} u\left(\theta - \tan^{-1}(p_x/p_y)\right) u\left(\psi - \tan^{-1}(h/p_y)\right),$$

(1)

where $\mathbf{s}_c = (s_x, s_y, s_z)$ is the Cartesian coordinate representation of $\mathbf{s}$ with $s_x = \rho(1 + \tan^2\theta + \tan^2\psi)^{-\frac{1}{2}}$, $s_y = s_x \tan\theta$, $s_z = s_x \tan\psi$, $\mathbf{n_p}$ is the unit normal vector to the point $\mathbf{p}$, $u(\cdot)$ is the unit step function, $\|\cdot\|_2$ is the $\ell_2$-norm of its vector argument, and $\angle(\cdot, \cdot)$ denotes the angle between its

vector arguments. The pair of step-functions in Eq. (1) model the occluding effects of the doorway wall.

Neglecting noise, background/visible side contributions, and other sources of model mismatch for the moment, the measured photograph of the visible ceiling is (approximately) proportional to the uniformly sampled radiance of the illuminated portion of the ceiling that is within the camera's field-of-view. The $m$-th camera pixel in an $M \times M$ (i.e., $M^2$ pixels) photograph, thus, measures an intensity value $y_m \propto \ell(\mathbf{p}_m, \mathbf{s})$ for each colour channel, due to a scene point $\mathbf{s}$. Here, $\mathbf{p}_m$ represents the spatial location on the ceiling plane sampled by the $m$-th pixel, while the constant of proportionality accounts for the projected size of a camera pixel on the ceiling plane and other global scalings introduced in the light detection pipeline of the camera (that are immaterial to the NLOS imaging task). Further, we choose to represent any hidden scene as a collection of angular surface elements, each with a unique discrete position in azimuth and projected-elevation angles and a continuous-valued range. In other words, we discretise along the two angular axes, but not along the range axis of our coordinate system, by dividing the hidden volume into $N_\theta$ equal azimuths (with separation $2\delta_\theta$), and $N_\psi$ equal projected-elevations (with separation $2\delta_\psi$). Within this discrete 2D angular grid, the $n$-th position $(\theta_n, \psi_n)$, with $n = 1, 2, \ldots, N_\theta N_\psi$, identifies the centre of a potential hidden scene surface element $\mathcal{S}_n = \{(\rho_n, \theta, \psi) : \theta \in [\theta_n - \delta_\theta, \theta_n + \delta_\theta], \text{ and } \psi \in [\psi_n - \delta_\psi, \psi_n + \delta_\psi]\}$ with range $\rho_n$. This gives a total of $N_\theta N_\psi$ contiguous, non-overlapping surface elements, where each surface element occupies an area of extent $[\theta_n - \delta_\theta, \theta_n + \delta_\theta]$ and $[\psi_n - \delta_\psi, \psi_n + \delta_\psi]$ along the azimuth and projected-elevation coordinates, respectively. A surface element $\mathcal{S}_n$, thus, subtends a solid angle (at the origin) whose size varies with angular position (Fig. 1c). Through this representation, an arbitrary scene is well-represented as a collection of all possible surface elements $\{\mathcal{S}_1, \mathcal{S}_2, \ldots, \mathcal{S}_{N_\theta N_\psi}\}$ with corresponding non-negative radiosities $\{f_1, f_2, \ldots, f_{N_\theta N_\psi}\}$ whose contribution to the $m$-th pixel measurement is:

$$y_m = \sum_{n=1}^{N_\theta N_\psi} f_n \int_{\mathbf{s} \in \mathcal{S}_n} \ell(\mathbf{p}_m; \rho, \theta, \psi) \delta(\rho - \rho_n) \frac{\rho^2 \sec^2(\theta) \sec^2(\psi)}{(1 + \tan^2(\theta) + \tan^2(\psi))^{\frac{3}{2}}} \, d\mathbf{s} \quad (2)$$

$$= \sum_{n=1}^{N_\theta N_\psi} f_n \int_{\theta_n - \delta_\theta}^{\theta_n + \delta_\theta} \int_{\psi_n - \delta_\psi}^{\psi_n + \delta_\psi} \ell(\mathbf{p}_m; \rho_n, \theta, \psi) \frac{\rho_n^2 \sec^2(\theta) \sec^2(\psi)}{(1 + \tan^2(\theta) + \tan^2(\psi))^{\frac{3}{2}}} \, d\psi \, d\theta, \quad (3)$$

where the unknown radiosity $c_n$ of the $n$-th surface element and the constant of proportionality are absorbed into $f_n$ without loss of generality. (We note that $f_n = 0$ indicates the absence of an object in the solid angle subtended by the elemental surface $\mathcal{S}_n$.) Collecting all scaled radiosities into an $N_\theta N_\psi$-dimensional column vector $\mathbf{f} = (f_1, f_2, \ldots, f_{N_\theta N_\psi})^\top$ and all measured pixel values into an $M^2$-dimensional column vector $\mathbf{y} = (y_1, y_2, \ldots, y_{M^2})^\top$ yields the discrete model $\mathbf{y} = \mathbf{A}(\boldsymbol{\rho})\mathbf{f} + \mathbf{v} + \mathbf{n}$, where the $N_\theta N_\psi$-dimensional column vector $\boldsymbol{\rho} = (\rho_1, \rho_2, \ldots, \rho_{N_\theta N_\psi})^\top$ contains the continuous-valued ranges for each surface element, and $\mathbf{A}(\boldsymbol{\rho}) \in \mathbb{R}^{M^2 \times N_\theta N_\psi}$ is a matrix whose entries follow from evaluating the integrals in Eq. (3), see equation (17) in Supplementary Note 1 for a closed-form expression. The vectors $\mathbf{v}$ and $\mathbf{n}$ model visible side background and measurement noise contributions, respectively.

## Reconstruction approach

Reconstructing the hidden scene from the measured colour photograph of light reaching the visible ceiling plane is equivalent to computing the configuration of the vectors $\mathbf{f}$ and $\boldsymbol{\rho}$ for the proposed surface element representation that best explains the photograph. Mathematically, from a measurement $\mathbf{y}$, we seek to recover $\mathbf{f}$ and $\boldsymbol{\rho}$

related by the discrete model $\mathbf{y} = \mathbf{A}(\boldsymbol{\rho})\mathbf{f} + \mathbf{v} + \mathbf{n}$. While the noise $\mathbf{n}$ is effectively handled by adopting a white Gaussian noise model, $\mathbf{v}$ is unknown. To resolve this, we assume that $\mathbf{v} = \mathbf{B}\mathbf{b}$ is slowly varying, and is approximately modelled as a weighted linear combination of an all-ones vector (representing visible side far-field contribution), and a near-field contribution $\mathbf{b}_{\text{vis}}$ that is due to a point source with a randomly chosen location on the visible side near the ceiling plane. Visible side background contributions are smooth and affect all pixels in the measurement plane, thus background contributions to the measurements are more likely to be explained by $\mathbf{B} = [\mathbf{1}, \mathbf{b}_{\text{vis}}]$ than $\mathbf{A}(\boldsymbol{\rho})$. Denoting the unknown weights of the two contributors in the background by $\mathbf{b} \in \mathbb{R}^2$, we formulate the following regularised inverse problem:

$$\arg \min_{(\mathbf{f}, \boldsymbol{\rho}, \mathbf{b}) \geq \mathbf{0}} \|\mathbf{y} - \mathbf{A}(\boldsymbol{\rho})\mathbf{f} - \mathbf{B}\mathbf{b}\|_2^2 + \mathcal{R}(\mathbf{f}), \quad (4)$$

where $\|\cdot\|_2$ denotes the $\ell_2$-norm, used here to ensure consistency of the estimates with the observation, and $\mathcal{R}(\mathbf{f})$ is a regulariser for $\mathbf{f}$ that is used to enforce sparsity or smoothness of estimates $\hat{\mathbf{f}}$. This optimisation in Eq. (4) is effectively tackled using the two-step approach illustrated in Fig. 2.

First, because the vertical and horizontal edges enable high azimuthal and projected-elevation resolution, respectively, and absent a similar mechanism for range resolution, we make the initial assumption that the entire hidden scene is confined to a fixed a priori chosen range $\rho_0$, and estimate the shape (angular elemental surface) representation of the hidden scene by solving the sparsity-constrained minimisation problem: $(\hat{\mathbf{f}}, \hat{\mathbf{b}}) = \arg \min_{\mathbf{f} \geq 0, \mathbf{b} \geq 0} \|\mathbf{y} - \mathbf{A}(\rho_0)\mathbf{f} - \mathbf{B}\mathbf{b}\|_2^2 + \lambda \|\mathbf{f}\|_1$, $\|\cdot\|_1$ denotes the $\ell_1$-norm regulariser, used here to enforce sparsity of the estimate $\hat{\mathbf{f}}$, and $\lambda$ is a non-negative scalar parameter controlling the trade-off between data consistency and sparsity. Solving the optimisation separately for each R-, G-, and B-colour channel in the measurement, we obtain corresponding estimates $\hat{\mathbf{f}}_R$, $\hat{\mathbf{f}}_G$, and $\hat{\mathbf{f}}_B$. Together these reconstructions yield a colour estimate of the projection of the hidden scene onto a surface of constant radius, with respect to the origin, defined by our projected-elevation spherical coordinate system. A 3D visualisation of this surface is shown in Supplementary Movie 2. Thus, as shown in Fig. 2b, c, visualising $\hat{\mathbf{f}}_R, \hat{\mathbf{f}}_G, \hat{\mathbf{f}}_B$ with $\rho_0$ in the proposed hidden scene representation reveals the shapes—from the vantage of the origin—of the hidden scene components placed at the same, and likely incorrect, range $\rho_0$. The first step concludes by identifying clusters of surface elements that likely constitute an object (see Fig. 2d). To that end, we form the binary vector $\mathbf{f}_{\text{bin}} = \mathbb{1}_{\left\{\hat{\mathbf{f}}_R + \hat{\mathbf{f}}_G + \hat{\mathbf{f}}_B > \gamma\right\}}$,

where the indicator function $\mathbb{1}_{\{\mathbf{x} > \gamma\}}$ returns a binary vector whose entries are one where corresponding entries of $\mathbf{x}$ are larger than $\gamma$ or are zero where corresponding entries of $\mathbf{x}$ are less than or equal to $\gamma$. We subsequently assign surface elements to the same cluster, if their corresponding non-zero entries in $\mathbf{f}_{\text{bin}}$ share an edge or vertex when it is viewed as a 2D photograph. In practice, we set a maximum number $J_{\max}$ of desired clusters, and order the recovered $J$ clusters based on their relative contributions to the measurement. The first $J_{\max} - 1$ clusters are retained while the rest are combined into a large super-cluster containing mostly spurious, possibly disjoint, background surface elements of low radiosities. We found this variation to be more robust in high noise and high visible side illumination scenarios.

The second step of the reconstruction approach aims to fit a new accurate range for each identified cluster, shown in Fig. 2d, by operating on greyscale mappings of the RGB observations and radiosities estimated in the first step. Assuming $J = J_{\max}$ clusters $\left\{\mathcal{C}_j\right\}_{j=1}^J$ are

computed, for each cluster $j$ we form a greyscale radiosity vector $\widehat{\mathbf{f}}_{grey}^{\mathcal{C}_j}$ by setting to zero entries in $\widehat{\mathbf{f}}_{grey} = (\widehat{\mathbf{f}}_R + \widehat{\mathbf{f}}_G + \widehat{\mathbf{f}}_B)$ corresponding to surface elements not present in cluster $\mathcal{C}_j$, with $j = 1, 2, \ldots, J$. (See the Supplementary Note 4 (S4.1) for more details on this step.) With their angular extents fixed, the clusters collectively produce a contribution to the measured photograph that depends non-linearly on their respective unknown ranges $\rho^{\mathcal{C}_j}$ and constituent surface elements (i.e. $\mathcal{S}_n \in \mathcal{C}_j$). The greyscale photograph $\mathbf{y}_{grey}$ computed as the sum of the three colour channels is related to the unknown cluster scalar ranges $\rho^{\mathcal{C}_j}$ and the estimated vector of radiosities $\widehat{\mathbf{f}}_{grey}^{\mathcal{C}_j}$ by:

$$
\begin{aligned}
y_{grey,m} = \sum_{j=1}^{J} w_j \sum_{n=1}^{N_\theta N_\psi} \hat{f}_{grey,n}^{\mathcal{C}_j} \int_{\psi_n - \delta_\psi}^{\psi_n + \delta_\psi} \int_{\theta_n - \delta_\theta}^{\theta_n + \delta_\theta} \ell(\mathbf{p}_m; \rho^{\mathcal{C}_j}, \theta, \psi) \\
\frac{(\rho^{\mathcal{C}_j})^2 \sec^2(\theta) \sec^2(\psi)}{(1 + \tan^2(\theta) + \tan^2(\psi))^{\frac{3}{2}}} \, \mathrm{d}\theta \, \mathrm{d}\psi + v_m + n_m,
\end{aligned}
\tag{5}
$$

with matrix-vector form $\mathbf{y}_{grey} = \mathbf{D}(\rho^{\mathcal{C}})\mathbf{w} + \mathbf{B}\mathbf{b} + \mathbf{n}$, where the weighting factors $w_j$'s allow the radiosity of each cluster to vary correspondingly when estimating their unknown continuous-valued ranges arranged into the $J$-dimensional vector $\rho^{\mathcal{C}} = (\rho^{\mathcal{C}_1}, \rho^{\mathcal{C}_2}, \ldots, \rho^{\mathcal{C}_J})^\top$, $\mathbf{w} = (w_1, w_2, \ldots, w_J)^\top$, and $\mathbf{D}(\rho^{\mathcal{C}}) \in \mathbb{R}^{M \times J}$ is a matrix whose $(m, j)$-entry models the contribution of cluster $j$ to measurement pixel $m$. The vectors $\mathbf{v}$, $\mathbf{n}$, $\mathbf{b}$ and matrix $\mathbf{B}$ are defined as before. We estimate the ranges of all $J$ clusters by solving the optimisation problem $(\widehat{\rho^{\mathcal{C}}}, \widehat{\mathbf{w}}, \widehat{\mathbf{b}}) = \arg\min_{(\rho^{\mathcal{C}}, \mathbf{w}, \mathbf{b}) \geq 0} \left\| \mathbf{y}_{grey} - \mathbf{D}(\rho^{\mathcal{C}})\mathbf{w} - \mathbf{B}\mathbf{b} \right\|_2^2$, using an accelerated projected gradient algorithm which is detailed in Supplementary Note 4 (S4.1). As confirmed by our FI analyses detailed in Supplementary Note 2, choosing to estimate a single range for each cluster $\mathcal{C}^j$ leads to a better-conditioned inverse problem because $J \ll N_\theta N_\psi$, compared to the alternative approach that attempts to recover a range for every surface element. Using the estimated ranges $\widehat{\rho^{\mathcal{C}}}$, visualised in Fig. 2e with the corresponding clusters, to construct the forward model $\mathbf{A}(\widehat{\rho^{\mathcal{C}}})$, the final 3D full-colour reconstruction (depicted in Fig. 2f) of the hidden scene is computed by solving the total variation regularised problem: $(\widehat{\mathbf{f}}^{TV}, \widehat{\mathbf{b}}^{TV}) = \arg\min_{\mathbf{f} \geq 0, \mathbf{b} \geq 0} \|\mathbf{y} - \mathbf{A}(\widehat{\rho^{\mathcal{C}}})\mathbf{f} - \mathbf{B}\mathbf{b}\|_2^2 + \lambda_{TV} \|\mathbf{f}\|_{TV}$ for each of the three colour channels. The total variation seminorm denoted as $\|\cdot\|_{TV}$ is used here to encourage piecewise smooth solutions, while $\lambda_{TV} > 0$ controls the trade-off between data consistency and piecewise smoothness. One may bypass the total-variation-constrained reconstruction and directly combine the recovered ranges with the initial full-colour shape reconstructions (in Fig. 2c) to obtain the fully 3D colour reconstruction of the hidden scene. This simplified variant of TERI which achieves impressive results at comparably lower computational complexity and three alternative reconstruction approaches are investigated in Supplementary Note 6. A narrated overview of the reconstruction procedure is provided in Supplementary Movie 1.

## Experimental reconstructions

Our reconstruction approach was assessed on several indoor scenes of varying complexities, containing multiple components with a variety of ranges, sizes, shapes, colours, rotation angles, and albedos. The hidden area was constructed from black foamboards framed using a rigid metal structure. These 1-in.-by-1-in. thick metal frames also formed the vertical and horizontal edges of the doorway, which deviates from the usual thin-edged occluders commonly used in some corner-camera-based demonstrations[28,33,39]. Several objects were then placed within the hidden region to obtain a variety of scenes.

The hidden scene was illuminated with an overhead light source attached adjacent to the hidden scene ceiling behind the doorway head. A digital camera focused on the visible ceiling plane was used to capture a photograph of the penumbra created on the ceiling plane. Figure 3 shows multiple views of the results of our reconstruction method for three example scenes. A collection of additional reconstructions for a variety of scenes and conditions is provided in Supplementary Notes 5 and 7.

Ground truth line-of-sight photographs of the hidden scenes are shown for visual comparison with their corresponding reconstructions. Approximate measurements of the ranges of hidden scene elements obtained by a laser distance metre (or tape measure) are reported in Table 1 along with corresponding range estimates for quantitative comparisons. Note that most scene objects occupy multiple ranges measured from the origin; we thus, report the ranges to the centres of each object. In each tested scene configuration, all objects constituting the hidden scene are accurately reconstructed, with visual inspection confirming correct shapes, positions, orientations, and colour reconstruction fidelity.

The backrest of the chair and the curvature of the basketball in Fig. 3b do not point toward the origin, hence, they deviate from our proposed projected-elevation spherical coordinate scene representation model. Regardless, they are recovered with great fidelity: The seat and backrest components of the chair are clearly identifiable, though the legs are not recovered since they are almost completely occluded by the seat from the measurement plane's perspective. Additionally, the backrest of the chair is dimmer than the seat. This is likely due to the backrest being perpendicular to the observation plane and nearly perpendicular to the hidden scene illumination.

The planar 'USF' characters (in Fig. 3c) are mostly legible. The ground truth 'U' which is stylised in a form resembling the horns of a bull has pointy tips that are even identifiable in the reconstruction. Observe also their recovered orientations. In particular, the yaw (rotation direction parallel to the ground plane) of the 'U', as well as the pitch of the 'SF' are also preserved in the reconstruction. However, the horizontal strokes of the 'F' are seemingly foreshortened in the reconstructions. This is explained by its deep placement (in azimuth) into the scene and orientation relative to the ceiling plane. This foreshortening effect is more exaggerated in the reconstruction of the celebrating mannequin standing on the ground (in the 'USF' scene): Its relative orientation to the ceiling plane is, therefore almost plan-view. Surprisingly, however, though much smaller, distant, and severely foreshortened, this fourth hidden scene component is correctly detected in the reconstruction. Its constituent colours are visible, and its range is accurately estimated.

Objects of varying range and size in the same scene are also evaluated (Fig. 3a). The pose, colour, and range of the small and close mannequin object are almost perfectly estimated; note the accuracy of its range estimate (reported in Table 1), which is likely due to its proximity to the origin. The colourful doughnut-shaped swimming tube (the largest and farthest hidden scene component) is recovered with surprisingly high fidelity, and relative sizes and positions of the different colours are well-preserved in the reconstruction. Notably, the large patches of blue on either side and the much smaller orange-coloured patches at the top and bottom of the doughnut are present. The reconstructed range and colours of the medium-sized and fairly distant volleyball object are comparably less accurate, likely due to the low reflectivity of the dark-blue stripes and distant spatial location.

Spanning the broadest continuum of range values from the origin, and composed of several large planar surfaces, many of which are orthogonal to the measurement surface, the shelf is by far the greatest exception to our hidden scene representation. However, the recovered shape is strikingly accurate when assimilated with care. First, we note that by being close to the origin, the top of the shelf occupies a larger subset of azimuthal angles compared to its farthest part (the base).

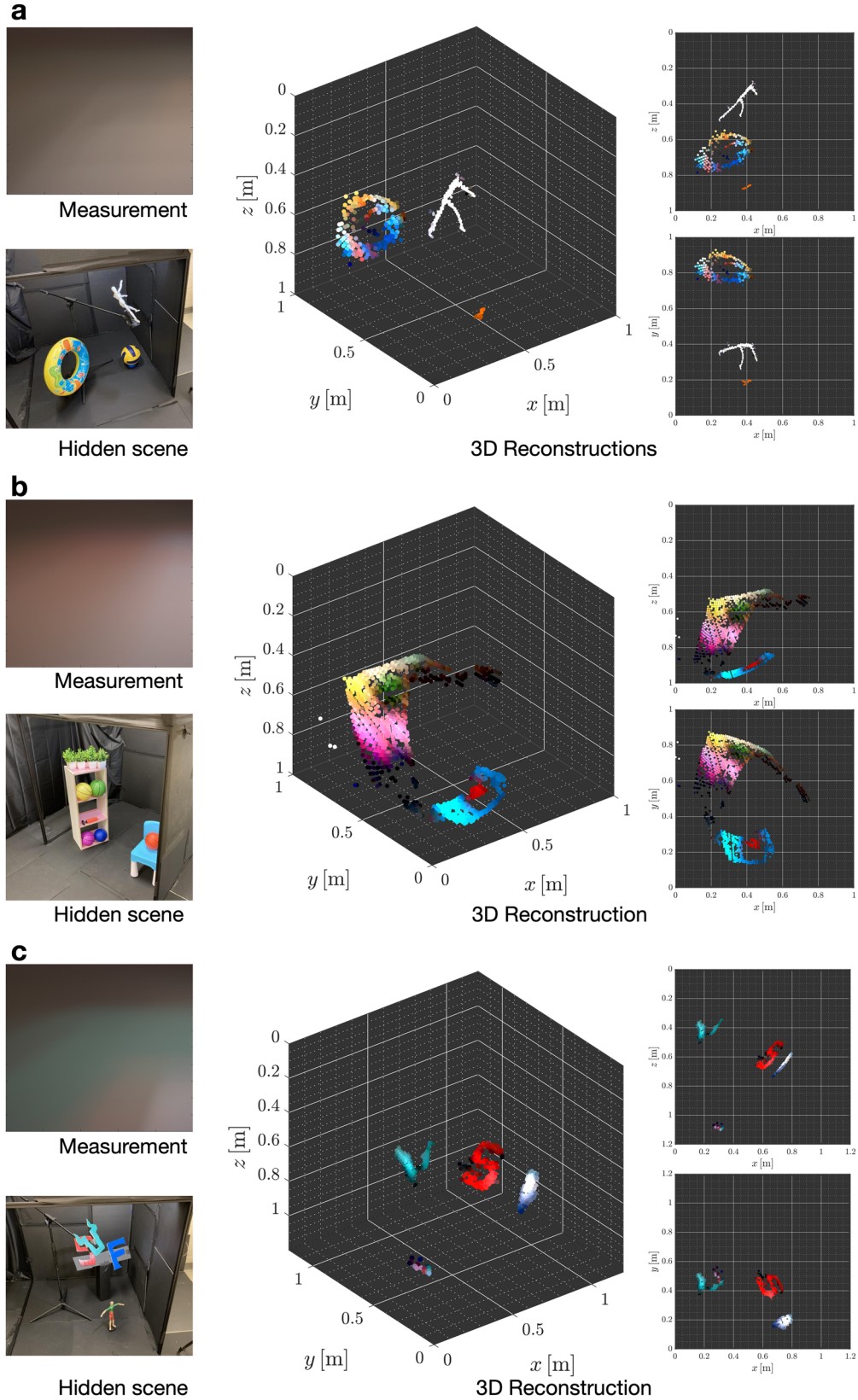

**Fig. 3 | Colour 3D reconstructions of three hidden scenes. a** Play scene. **b** Work scene. **c** USF scene. For each scene, the first column shows the NLOS measurement photograph (top), and a line-of-sight photograph of the true scene (bottom); the second column shows the 3D full-colour reconstruction; and the last column shows the side (top) and plan (bottom) views of the 3D reconstruction. The reconstructions shown here were obtained with $J_{max} = 4$ for (**a**) and (**b**), and $J_{max} = 5$ for (**c**).

**Table 1 | Experimentally measured and estimated ranges of hidden scene components shown in Fig. 3**

| Scene | Object | Measured range [m] | Estimated range [m] |
|---|---|---|---|
| | Doughnut | 1.32 | 1.10 |
| Play scene | Mannequin | 0.58 | 0.65 |
| | Volleyball | 1.14 | 0.97 |
| | Shelf | 1.12 | 1.04 |
| Work scene | Basketball | 0.99 | 1.03 |
| | Chair | 1.02 | 1.03 |
| | U | 0.64 | 0.63 |
| USF scene | S | 1.12 | 0.99 |
| | F | 1.07 | 1.01 |
| | Mannequin | 1.09 | 1.23 |

The average range reconstruction error is roughly 9.3 cm, demonstrating surprisingly accurate range estimation.

Most objects on the shelves also appear in the reconstructions. Looking closely, near the top of the shelf are two neighbouring patches of yellow (left) and green (right) representing the two basketballs in the topmost cubby. Following beneath is a vibrant pink area corresponding to the surface of the mostly empty second cubby hole. The final hole shows subtle patches corresponding to the pink and dark blue basketballs. Because the shelf is so detailed with different colours, the total variation prior smears out smaller details, such as the objects on top of the shelf. However, in Supplementary Note 4, we provide a less smoothed reconstruction performed without the total variation constraint (see Supplementary Fig. 13) that shows more details of the shelf.

Generally, ambient light sources illuminating the visible area will attenuate the informative penumbra, therefore decreasing the fidelity of our measured photographs and making accurate reconstructions even more challenging. We evaluate the limits of our approach in this scenario by introducing increasing levels of visible side illumination. Results of our evaluation are summarised in Fig. 4. Overall reconstruction performance degrades uniformly with increasing illumination strength. The reconstruction in medium ambient light (Fig. 4c) has a large smeared artefact (likely attempting to fit the extraneous background contributions). Generally, despite the discolorations and erroneous clutter introduced, the reconstructions retain impressively accurate visual information about the scene: even as the penumbra becomes imperceptible by the human eye (Fig. 4c, d), the reconstruction is still recognisable as having two objects with fairly accurate poses and positions. Specifically, the red-green mannequin in the T-pose appears with surprisingly accurate colour, however, the white mannequin appears fainter and discoloured. The beige-coloured arms and legs of the red-green T-pose mannequin are visible and distinguishable from the red and green parts of the mannequin in all ambient illumination levels. Additional experiments to evaluate the range estimation accuracy of the system are presented in Supplementary Note 5 (see Supplementary Table 1).

Our approach and demonstration of accurate full-colour 3D imaging of hidden scenes from ordinary photographs of penumbra formed on a visible ceiling surface represent a significant advance in passive NLOS imaging. Prior to our work, intensity-based passive approaches were limited to computing one- or two-dimensional plan view reconstructions of small-scale scenes, with coarse range estimates. Here, TERI achieves high-resolution 3D reconstructions of medium- to large-scale scenes by exploiting a two-edge acquisition configuration and an efficient and accurate reconstruction approach. The approach is inspired by a theoretically-motivated and computationally efficient hidden scene representation. Our reconstruction resolution and quality are comparable to active 3D NLOS methods while using substantially cheaper and more ubiquitous equipment (that does not require a specialist operator). In addition, our hidden scene representation using elemental surfaces of continuous-valued ranges is more memory-efficient than complete hidden scene voxelisation, which also discretises the range coordinate and therefore scales cubically with the desired scene reconstruction resolution.

## Discussion

Despite the successful demonstrations of TERI shown here, certain aspects could be improved. Our modelling and reconstruction approaches do not incorporate self-occlusion among multiple hidden scene objects. This may cause penumbrae in the measurements that are not explainable by the forward model, and thus could potentially be a significant source of model mismatch, depending on the observation FOV. However, our closed-form expressions for hidden scene elements could facilitate efficient modelling of self-occlusions in the style of ref. 39 and the development of an accompanying reconstruction technique.

Moreover, extending TERI to allow the estimation of multiple ranges per cluster, instead of confining them to a single range, could produce better reconstructions and visualisations for scenarios with objects that span a large continuum of ranges. A suitable deprojection postprocessing step could map the estimated projected-elevation spherical coordinates surface elements to planar surfaces in Cartesian coordinates; one would also need to compute the orientations of the deprojection planes for each object. An alternative approach is to recursively segment the object into smaller portions in a multi-resolution approach while simultaneously recomputing a range estimate for each sub-segment. Additionally, when objects positioned at different ranges occupy overlapping or contiguous solid angles, our current algorithm may incorrectly reconstruct them as being at one range. This results in possibly biased range and radiosity estimates for all objects. Preliminary demonstrations presented in Supplementary Note 7 suggest that a recursive segmentation scheme can feasibly alleviate incorrect clustering of surface elements, hence opening up an opportunity for more thorough future explorations. Nevertheless, our successful use of a simple algorithm to identify clusters of surface elements was motivated by simplicity and low computational complexity while also being effective for various scenarios. A more sophisticated clustering approach, based on cluster compactness, smoothness, colours, local pixel similarities, or mixtures thereof, or even a trained neural network approach, could yield improvements in clustering accuracy and further facilitate the removal of spurious surface elements arising from noise, model mismatch, and visible-side contributions.

TERI could also be combined with other active techniques that similarly exploit edges[28,39]. In particular, our acquisition approach and modelling could also be incorporated into the recent active corner camera[28] for fully 3D object reconstructions and simultaneous tracking and scene mapping. In general, active approaches demonstrate significant robustness under ambient illumination; thus, a synergistic combination may enable a technique that has reduced acquisition times relative to active NLOS imaging techniques while also being more robust to ambient light contributions.

Our reported experiments focussed on reconstructing a single octant, i.e. the recoverable portion of the room contained within $\theta \in [0, \pi/2]$ and $\psi \in [0, \pi/2]$. However, extending the approach to additionally image a second octant, wherein the entire recoverable portion of the room is $\theta \in [0, \pi]$, is possible with fairly minor algorithmic modifications. The developed representation model and theory still apply in both cases. However, the thickness of the occluding wall would need to be modelled. Furthermore, we considered scenes that fit inside a roughly $1.2 \times 1.2 \times 1.2$ m³ room. However, we speculate that the current setup would succeed for typical rooms of around

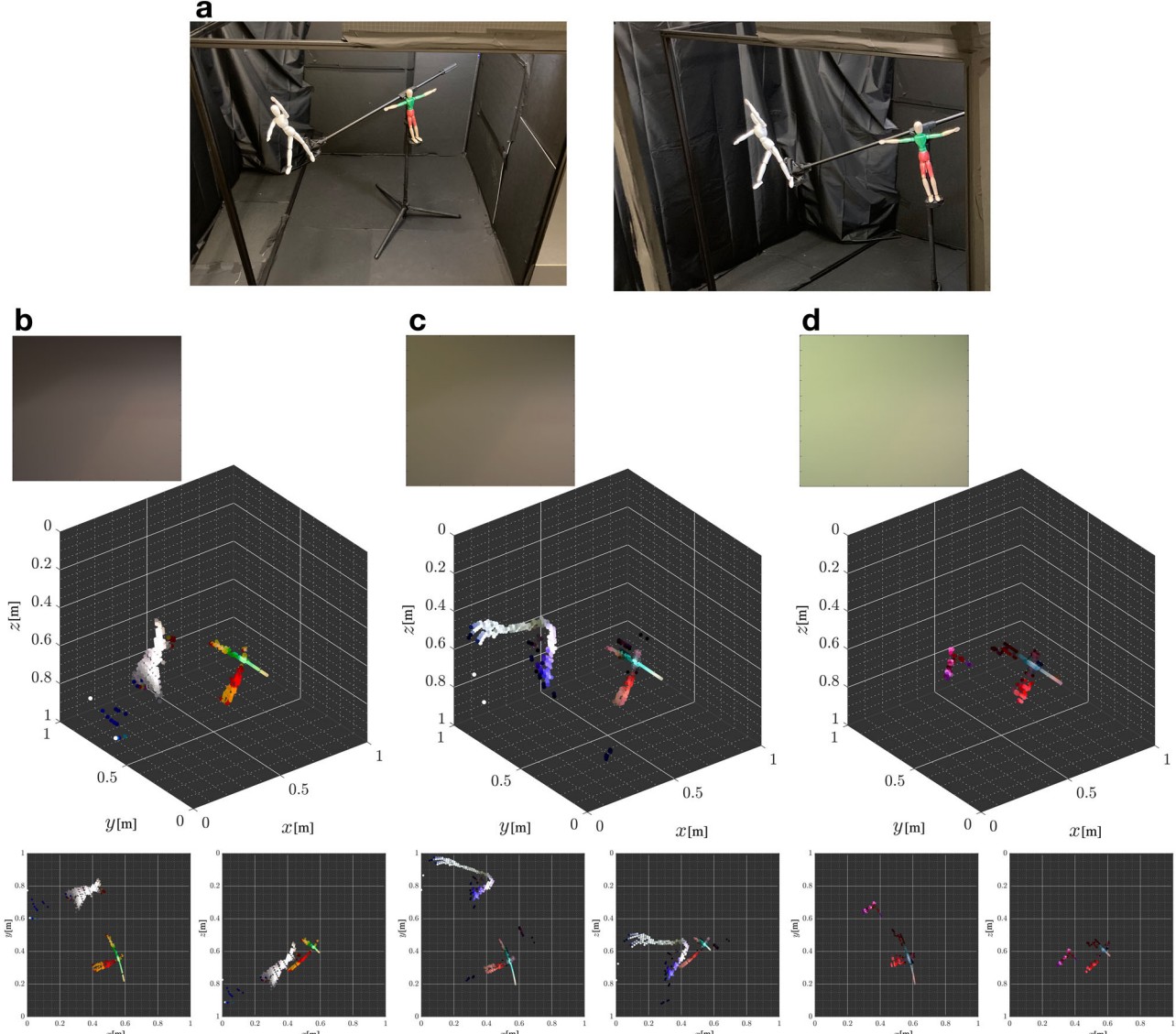

**Fig. 4 | Reconstructions under increasing levels of visible side illumination. a** Photograph of the hidden scene with two mannequins striking different poses: A red-green mannequin in a T pose and a white mannequin in a marching pose measured at 0.86 and 1.04 m, respectively, from the origin. **b** With visible side illumination turned off, the ranges of the mannequins are estimated to be 0.86 m for the T pose mannequin and 1.10 m for the marching mannequin. **c** Moderate amount of visible side illumination introduced; here, the ranges of the mannequins are to be 0.85 m for the T pose mannequin and 1.10 m for the marching mannequin. **d** High amount of visible side illumination introduced; the T pose mannequin is reconstructed as two disjoint clusters of elemental surfaces (representing upper and lower halves of the mannequin) with estimated ranges of 0.90 m and 0.92 m, and the marching mannequin is estimated to be at 0.98 m from the origin. The reconstructions shown here were obtained with $J_{max} = 3$. Best viewed in colour.

$3 \times 3 \times 2\,m^3$ containing human-sized occupants, with little degradation in performance (especially when there is little to no additional visible side ambient illumination and the room walls are minimally reflective). We base this on the surprising reconstructions obtained in ambient illumination shown in Fig. 4, as well as the detection and fairly accurate ranging of the small volleyball in Fig. 3a. These reconstructions are computed from low-signal situations, as would be similarly observed in larger-scale scenes. Indeed, as the actual scene sizes increase further, accurate imaging typically becomes harder because (i) the irradiance of the illuminated ceiling patch caused by the hidden scene objects reduces, becoming nearly constant at far-field, and this makes ranging harder; and (ii) the produced penumbra becomes less pronounced, making shape reconstruction more challenging. Using more intense illumination of the hidden scene, detectors with greater sensitivity, dynamic range, resolution, and lower noise floor, and increasing the measurement FOV are possible mechanisms for enabling accurate reconstructions in these more challenging conditions.

When scenes have large, flat, and highly reflective backgrounds—such as a room with a white floor and white walls—our algorithm produces 3D reconstructions that accurately indicate all solid angles containing reflective surface elements and can even reveal hidden scene shadows (see additional experiments presented in Supplementary Figs. 24 and 25). Despite this, it is challenging to intuit the scene configuration from the 3D visualisations, especially when a foreground object exists and takes the same colour as the background walls. In this scenario, our clustering algorithm cannot distinguish foreground surface elements from background surface elements because they are angularly contiguous. Contributions from such large, planar surfaces may also dominate informative penumbra cast by smaller hidden-scene components.

## Methods

### Equipment

The observation photographs were obtained using a Kiralux® CMOS compact scientific camera with 2048 × 2448 resolution (5 Megapixels) by Thorlabs (part number: CS505CU), equipped with a 25 mm, $f$/1.8 aperture, 2/3 in. format lens (Thorlabs part number: MVL25TM23). ThorLabs' ThorCam™ Software for Scientific and Compact USB Cameras installed on an MSI 2019 laptop (Intel Core i7-10510U with 32 GB memory) computer was used to control the camera. Data processing and simulations were performed using the laptop above and, additionally, on an Apple MacBook Pro 2017 (3.1 GHz Quad-Core Intel Core i7 with 16 GB memory) with MATLAB. Twelve 48-inch-long structural rails each with a thickness of 1-in.-by-1-in. (Thorlabs part number: XE25L48) were used to create the cubic hidden scene (length 48 in.). The floor and sides were covered using a combination of black foam boards and black weaved nylon fabric with polyurethane coating (Thorlabs part number: BK5). To create the ceiling plane, white foam bards were supported ~0.29 m above the top face of the cube using black foamboards (to form the doorway head). This created a hidden scene area whose height was 1.51 m above the ground, with a square base of length 1.22 m. The structural rails thus formed both edges of the occluding doorway walls.

### Experimental details and observation pre-processing

The camera, positioned at a height of 0.4572 m above the floor plane using a tripod stand, was pointed directly upwards and focussed on the ceiling plane. Hence, the distance between the camera's sensor plane and the observation surface was roughly 1.07 m. The camera orientation was such that the lower limit of its FOV of the visible part of the ceiling plane was coincident with the intersection between the ceiling and the wall separating the visible and hidden regions, while the right boundary of the camera's FOV aligned with the vertical edge of the occluding wall. This made the bottom right corner of the captured photograph the origin of the $xy$-plane. Although the camera has a maximum exposure time of 7331 ms, for each tested hidden scene configuration, the exposure time used was manually adjusted to avoid saturation while approximately using the full dynamic range of the camera. For reduced measurement noise, 50 snapshots were taken and averaged to obtain a single photograph. Each photograph was then resized and cropped to have an equal number of horizontal and vertical pixels and was subsequently downsampled in preprocessing using a median filter to produce a final 125 × 125-pixel colour (i.e., RBG) photograph, over a FOV of size 0.46 m-by-0.46 m. Additional visible side ambient light was introduced for the two-mannequins test scene, by using the flashlight of a smartphone (Apple iPhone 8) at increasing levels to illuminate white foam boards arranged vertically and horizontally, as well as the white floor and wall surfaces in the visible side of the room (see Supplementary Fig. 16).

## Data availability

The raw penumbra measurement data used to produce the results reported in the manuscript and supplementary information (Figs. 3 and 4, Table 1, as well as Supplementary Figs. 6, 10, 11, 12, and Supplementary Table 1) are available on GitHub at https://github.com/iscilab2020/TERI-3DNLOS, and Zenodo[44].

## Code availability

Code used to produce all experimental and simulation-based figures in the manuscript and supplementary information (Figs. 3 and 4, Table 1, as well as Supplementary Figs. 6, 10, 11, 12, and Supplementary Table 1) are available on GitHub at https://github.com/iscilab2020/TERI-3DNLOS, and Zenodo[44].

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

## Acknowledgements
The authors thank V. K. Goyal, H. Joudeh, C. P. Saunders, and S. W. Seidel for discussions and feedback on an initial version of the manuscript.

## Author contributions
J.M.-B. conceived the project and supervised the research. R.C. conceived the acquisition method and scene representation model, derived the light transport model, and performed numerical simulations. R.C. and J.M.-B. developed the reconstruction algorithms, designed and performed the experiments, discussed and analysed the data, and wrote the manuscript.

## Competing interests
The authors declare no competing interests.
