## [Peer Review File · Nature Communications]

Two-edge-resolved three-dimensional non-line-of-sight imaging with an ordinary cameraREVIEWER COMMENTS

Reviewer #1 (Remarks to the Author):

The work by Czajkowski and Murray-Bruce reports on the development a new approach to non line of sight imaging that applies to passively illuminated scenes and is able to reconstruct RGB images that also include depth information. This is achieved by implementing a multistep inverse retrieval approach that in turn relies on a forward model for predicting the shadows cast by hidden objects, when viewed on a ceiling close to the two-edge profile of a door frame.

NLOS imaging has come a long way since the proposals and demonstration just over 10 years ago. This work shows just how far. One could argue that the scene reconstructions are not perfect and I suspect that if one was not shown the ground truth images (that allow the brain to implicitly interpolate the reconstruction results and make sense of them) it might actually be hard, based on the reconstructions alone, to formulate a definite opinion about what the hidden objects actually are. However, I still find it quite impressive that the authors are able to reconstruct 3D depth, in colour of relatively complicated scenes (i.e. with multiple objects). The colour renditions are not always perfect and shapes are sometimes not obvious, but the simple ability to locate the position and rough shape of objects with so little information, i.e. just one single colour photograph of the shadows cast close to a door frame, is quite impressive.

The manuscript is very well prepared and well presented. I do have some minor points that the authors might want to consider:

- 1) I am not sure that I fully understand figures 1d and 1e and they are not really helping me understand what point the authors are trying to make
- 2) line 265: the authors describe their clustering algorithm in terms of the grouping of elements depending on whether they are adjacent in a 2D projection image. So does mean that any objects that are separated in depth but overlap in the 2D projection will be clustered and therefore "forced" to be clustered at a single depth (thus also resulting in a change in luminosity of one or both objects)? If so, it might be worth pointing this out explicitly
- 3) Figure 3 and in general all figures - is it possible to put units on the axes? This would be very helpful in understanding the scales and overall dimensions. Related to this point - the door edge thickness is 1", but what is the size of the doorway aperture?
- 4) similar point, related to figures in SM, e.g. section 5 - it is good to see photos of the overall scene with labels. However, it is not clear looking at these where the ceiling surface is located from which the RGB photo is actually taken. Is it possible to visualise/indicate this too?
- 5) On a similar note - the current scene volumes seem to be of order $1 \times 1 \times 1 \text{ m}^3$, is that correct? Can the authors speculate on how large these scene volumes could become? If one wanted to upscale to say $3 \times 3 \times 2 \text{ m}^2$ (e.g. maybe a typical room size), is it just a question of ensuring that there is sufficient light (more intense illumination or more sensitive camera + multiple exposures etc) or does the model tell us that the shadows/projections become more diffuse, less defined, therefore making the reconstructions harder and less precise? More in general, is there an indication that the precision with which scenes are reconstructed, varies with the size of the actual scene? It is clear from everyday experience that as the "edges" are moved away from objects in a scene, these become smoother and features in the shadows tend to blur out. So I would guess that small scene volumes are easier to reconstruct compared to larger ones? A discussion along these lines would be very helpful (and I do not think would detract from the importance of this work even if there are limitations in total volumes that can be imaged)

Overall, I enjoyed reading this manuscript, it is very high quality work, it is novel and does represent a significant step forward in the NLOS imaging field.

I would therefore recommend publication as is or ideally, with some minor modifications following the suggestions above assuming that all of these make sense.

Reviewer #2 (Remarks to the Author):

Key Results:

This paper by Czajkowski and Murray-Bruce presents the first passive NLOS method to produce compelling 3D results. The method requires two key innovations and some reasonable signal processing assumptions. The first major innovation is the use of doorframe as an occluder that is sufficiently ubiquitous in indoor settings to be useful, and which crucially enables high angular resolution in a second dimension, compared to existing "corner camera" approaches. Secondly, the authors propose a novel coordinate system that yields critical separability in the two angular dimensions. This coordinate system description allows the authors to derive the light propagation model in a simplified way that facilitates more straightforward inversion. Taking advantage of doorframe occluders that exist "in the wild," the two-edge-resolved NLOS imaging configuration is able to reproduce full color 3D images of macroscopic scenes with consumer-grade equipment and reasonably high accuracy.

Validity:

The data acquisition and processing approaches are largely valid. I have some suggestions below that may lead to slightly better results, but the approach here is principled and reasonable.

Significance:

These results are a significant step forward for the field. Current assumptions are that high-quality 3D NLOS results require expensive, specialized equipment. This paper demonstrates that there is significant 3D information that can be extracted from only a standard camera.

Clarity and context, References:

The paper is clearly written and properly cites the relevant contributions in the literature.

Data and methodology, analytical approach:

A detailed discussion of the datasets and methodology are discussed below.

Acquisition strategy

- Like most NLOS demonstrations, the setting is somewhat contrived, with a few objects of interest within a minimally reflective background and with a doorframe opening. However, using multiple (orthogonal) edges of a doorframe is a clever approach to passive NLOS with some clear advantages of the two-edge-occluder camera. First, the head of the doorframe naturally blocks light sources on the ceiling, which are a major source of interference in other passive settings. Furthermore, although other work with 2D or 3D occluders have shown the best results for passive NLOS imaging, the occluder shape either must be known or estimated. Having an (approximately 2D) occluder that is visible to the observer is both less contrived and also eliminates one of the challenges of existing passive methods.

Scene representation model

- It took reading the supplementary material in detail to understand why the projected elevation coordinate system makes such a large difference in the reconstruction. Perhaps more intuition can be given for why this system is a natural fit for TERI and significantly better than spherical coordinates. If one considers a single edge occluder, then cylindrical coordinates are a natural fit, as have been explored in several papers. Using two orthogonal edge occluders effectively yields a pair of single edge cameras with complementary information in orthogonal directions. (Can one think of these coordinates as somehow a compromise between two complementary cylindrical coordinate systems?) The projected elevation system allows one to describe angular coordinates

from each of the edge occluders separately, whereas using spherical coordinates forces the elevation description to depend on the azimuth.

- This separation greatly simplifies the forward modeling, which I assume makes the reconstruction problem much easier. There is a hint to the advantage in the mention of the information orthogonality, which seems fascinating and potentially critical to actually being able to solve the inverse problem, but more intuition/explanation should be given to explain how that makes a meaningful difference. (i.e., now that we see the angular coordinates are uncoupled in the new coordinate system, what are the downstream effects for forward model/inverse problem?)

Reconstruction method

- The approach of solving for shape first and then range is interesting. I'm surprised that initialization with a single depth is good enough to recover shape reasonably well. Are there any constraints on how the initial depth is selected?

- After recovering the shape, estimating the range is the more difficult problem to solve, although there is some noticeable improvement based on improving the range estimate. I wonder whether there would be any benefit in alternating between the shape/range estimation procedures. Could the shape be refined from repeating that step even once after the range has been corrected?

- The assumption of all points in a cluster being located at the same depth irks me a bit. I understand the authors have tried to justify this by showing that estimating a range for every point in the scene would have far too little information for robust estimation. Figure S8 in the supplement justifies that the CRB is higher for more clusters. However, the CRB is a lower bound on the variance of an *unbiased* estimator. I would argue that assuming all points in a cluster are located at a fixed range is a biased estimator, and subsequently breaking a cluster into progressively smaller subclusters allows for lower bias but higher variance. With no clusters at all (independent range estimate per pixel), the mean-squared error is understandably far too high. However, it's not obvious that for every scene the high-bias, low-variance estimate of a single cluster will yield lower MSE than the lower-bias higher-variance estimate of a small number of clusters. This is especially true for extended objects such as the shelf, which is visibly distorted in Fig. 3b. I wonder whether a slightly more sophisticated method would decide on clusters based not only on connected components, but also how compact those components are in 3D. So after the 2 steps (shape/range estimation) in the existing algorithm, one could check whether the single-cluster assumption seems reasonable. Using the single-cluster estimate as the initialization, smaller subclusters could then refine the 3d position estimate for more realistic results.

- The tested scenes, with a small number of colorful objects surrounded by a black background, encourage the use of the sparsity prior (i.e., l_1 -norm regularization) for reconstruction. What are the limitations of this assumption? Would it be possible to reconstruct an empty room with white walls, which would reflect light from all angles? What about a white object in front of a white wall, or more generally any object partially occluding an object of the same color?

- I don't quite understand the background light modelling. For the near-field contribution from a point source on the visible side, is it assumed the point source location is known? Or does b_{vis} allow for a point source at any location (which might collapse the linear independence with the all-ones vector)?

- For the reconstruction display, the colors seem skewed relative to the true photographs. In particular, yellow objects appear to be far more red than the true objects. I think the problem is that the raw RGB values are being mapped directly to the 3 color channels of the point cloud display. I wonder whether first multiplying by a color conversion matrix (see doi:10.1117/1.OE.59.11.110801) and/or white balancing would yield more perceptually consistent results.

- How are the threshold γ and number of clusters J_{max} chosen?

Additional Comments

- I find Fig. 1b to be hard to follow. I understand the idea of the projected elevation, but something about the 3D structure projected onto 2D of the page is hard to interpret. Perhaps using a wireframe cube to show the 3d position in the Cartesian coordinates would help make the projection of the point onto the xz plane easier to interpret.

- How is the projected elevation coordinate related to the classical elevation coordinate?

- Lines 38-41: Not clear whether this is still referring to ref. 27.

- Figure 2 caption: "Four such clusters were identified, one for each object" → "one for each of the three objects"

- Line 316: In what way is the chair + basketball an exception to the modeling?

- Several of the references on arxiv have appeared now in journals/conferences

- Supplement Fig. 3: what does the coloring of the surface element represent?

- Supp Fig. 6:

o subfigures not labelled

o The improvement in CRB with more edges is remarkable! Very illustrative result

- Perhaps I missed it, but what is the δ in Supp. Eqn. (41)?

- Supp. Fig. 14: Some colored text markup should be removed.

Response to the Associate Editor and Reviewers of
NCOMMS-23-28870

**Two-edge-resolved three-dimensional
non-line-of-sight imaging with an ordinary camera**

R. Czajkowski and J. Murray-Bruce

November 9, 2023

Intro Statement

We thank the Associate Editor and both reviewers for their thoughtful comments and encouragingly positive feedback on our manuscript. These comments have enabled us to further improve the quality of our work.

Based on the feedback, we have successfully improved the accuracy of colour renditions achieved in our work (see Figures 2, 3, and 4 of the main manuscript), the clarity of concepts conveyed by Figure 1, discussed the downstream effects of the proposed modelling, and further discussed the limitations of our work. We are thrilled to showcase our modifications and additions here and to share this work with the community.

New experiments and reconstructions are included in this response document. Some of those reconstructions made it into the main manuscript. Most others were included in the supplementary document (and referenced in the main manuscript), since they will certainly benefit technical readers and aid those who may have similar questions to the reviewers. Our choices on components that made the main manuscript aimed to balance technical nuance with simplicity in the main manuscript. We hope the reviewers will agree with our decisions.

For convenience, we coloured changes made to the main manuscript as markup. We also made the following changes (not marked up) to the supplementary document:

- Updated the TERI algorithm to incorporate a smoothness prior in producing the final reconstruction (Supplementary Section S4; pp. 23)
- Included previous version of our reconstruction algorithm in the supplement as ‘Variant of TERI algorithm I: No total variation refinement’ (Supplementary Section S4.2; pp. 24)
- Included a new, clearer, labelled photograph of the experimental setup (Supplementary Figure 16; pp. 31)
- Updated Supplementary Figure 17 based on the new TERI algorithm (previously Supplementary Figure 16; pp. 32)
- Added Supplementary Section S5.5 on reconstructing rooms with highly reflective walls (pp. 39)
- Added Supplementary Section S7 to investigate the feasibility of refining large clusters (pp. 52)

We again would like to thank both reviewers for taking the time to provide critical and detailed reviews of this work.

1 Response to Reviewer #1

Comment 1.1

The work by Czajkowski and Murray-Bruce reports on the development a new approach to non line of sight imaging that applies to passively illuminated scenes and is able to reconstruct RGB images that also include depth information. This is achieved by implementing a multistep inverse retrieval approach that in turn relies on a forward model for predicting the shadows cast by hidden objects, when viewed on a ceiling close to the two-edge profile of a door frame.

Response:

This is an accurate summary of our work. Thank you for reviewing our work.

Comment 1.2

NLOS imaging has come a long way since the proposals and demonstration just over 10 years ago. This work shows just how far. One could argue that the scene reconstructions are not perfect and I suspect that if one was not shown the ground truth images (that allow the brain to implicitly interpolate the reconstruction results and make sense of them) it might actually be hard, based on the reconstructions alone, to formulate a definite opinion about what the hidden objects actually are. However, I still find it quite impressive that the authors are able to reconstruct 3D depth, in colour of relatively complicated scenes (i.e. with multiple objects). The colour renditions are not always perfect and shapes are sometimes not obvious, but the simple ability to locate the position and rough shape of objects with so little information, i.e. just one single colour photograph of the shadows cast close to a door frame, is quite impressive.

Response:

Thank you for the positive comment. We were also pleasantly surprised by the level of detail in shape, and the accuracy in range estimation. In addition, we believe we have been able to improve on the colour renditions through the use of a TV prior explained below.

Comment 1.3

The manuscript is very well prepared and well presented. I do have some minor points that the authors might want to consider: 1) I am not sure that I fully understand figures 1d and 1e and they are not really helping me understand what point the authors are trying to make.

Response:

Thanks for the positive comment on the clarity of our manuscript. We agree with the reviewer's comment that the initial version of Figures 1d and 1e were not as effective as we intended. With those figures, we aimed to illustrate an intuition for our proposed coordinate system and the mechanism that provides 2D angular resolution. While we found difficulty doing so via 2D figures, we believe other parts of our manuscript communicated these concepts – especially given your accurate summary of our work.

Reviewer 2 expressed difficulty in interpreting Figure 1b and had a great suggestion for it. We adapted their suggestion to similarly improve Figures 1d and 1e. We hope our ideas are better communicated in the updated version of Figure 1. In addition, Supplementary Video 2 provides an animation of the concepts.

Comment 1.4

2) line 265: the authors describe their clustering algorithm in terms of the grouping of elements depending on whether they are adjacent in a 2D projection image. So does mean that any objects that are separated in depth but overlap in the 2D projection will be clustered and therefore "forced" to be clustered at a single depth (thus also resulting in a change in luminosity of one or both objects)? If so, it might be worth pointing this out explicitly.

Response:

We agree that this weakness of our algorithm ought to be made more explicit in our manuscript. We have added the line to the manuscript:

Additionally, when objects positioned at different ranges occupy overlapping or contiguous solid angles, our current algorithm may incorrectly reconstruct them as being at one range. This results in possibly biased range and radiosity estimates for all objects. Preliminary demonstrations presented in the supplementary document suggest that a recursive segmentation clustering scheme can feasibly alleviate incorrect clustering of surface elements, hence opening up an opportunity for more thorough future explorations. Nevertheless, our successful use of a simple algorithm to identify clusters of surface elements was motivated by simplicity and low computational complexity while also being effective for various scenarios. A more sophisticated clustering approach, based on cluster compactness, smoothness, colours, local pixel similarities, or mixtures thereof, or even a trained neural network approach, could yield improvements in clustering accuracy and further facilitate the removal of spurious surface elements arising from noise, model mismatch, and visible side contributions.

Additionally, Reviewer 2 also commented on the shortcomings of the clustering algorithm (see Comment 2.10). Both comments inspired additional experiments and a clustering algorithm that further refines a large cluster by dividing it up into smaller constituent sub-clusters. In the current demonstration, the subdivisions are along or across the centroid of a cluster. The resulting reconstructions are encouraging and suggest that a more sophisticated algorithm that applies to a broader range of situations is feasible. We intend to explore this in a follow-up work. The experiments and algorithmic details of the feasibility study are included in the supplementary document (Supplementary Section S7).

Comment 1.5

3) Figure 3 and in general all figures - is it possible to put units on the axes? This would be very helpful in understanding the scales and overall dimensions. Related to this point - the door edge thickness is 1", but what is the size of the doorway aperture?

Response:

Figures 2, 3, and 4 of the main manuscript now have units. Thanks for drawing our attention to this. The doorway aperture is $1.2 \times 1.2 \text{ m}^2$. The following figure, which appears as Supplementary Figure 16, shows the doorway aperture.

Supplementary Figure 16: An annotated photograph of the experimental setup. Here, the inset shows the view from the camera's perspective, which is of a white foamboard that plays the role of the ceiling.

Comment 1.6

4) similar point, related to figures in SM, e.g. section 5 - it is good to see photos of the overall scene with labels. However, it is not clear looking at these where the ceiling surface is located from which the RGB photo is actually taken. Is it possible to visualise/indicate this too?

Response:

Great suggestion! We believe the figure accompanying our response to your previous comment (Comment 1.5 immediately above) fills this need.

Comment 1.7

5) On a similar note - the current scene volumes seem to be of order $1 \times 1 \times 1 \text{ m}^3$, is that correct? Can the authors speculate on how large these scene volumes could become? If one wanted to upscale to say $3 \times 3 \times 2 \text{ m}^3$ (e.g. maybe a typical room size), is it just a question of ensuring that there is sufficient light (more intense illumination or more sensitive camera + multiple exposures etc) or does the model tell us that the shadows/projections become more diffuse, less defined, therefore making the reconstructions harder and less precise? More in general, is there an indication that the precision with which scenes are reconstructed, varies with the size of the actual scene? It is clear from everyday experience that as the "edges" are moved away from objects in a scene, these become smoother and features in the shadows tend to blur out. So I would guess that small scene volumes are easier to reconstruct compared to larger ones? A discussion along these lines would be very helpful (and I do not think would detract from the importance of this work even if there are limitations in total volumes that can be imaged)

Response:

That's correct, current scenes fit inside of the roughly $1.2 \times 1.2 \times 1.2 \text{ m}^3$ room. This question is very interesting with some nuance. We will try our best to address it concisely.

As the scene size increases we highlight some expected observations. In the case where object depths increase but object sizes are held fixed, one would observe that:

- The perceived brightness of the object from the ceiling's perspective reduces due to radial falloff, and the illumination/irradiance of the ceiling patch caused by the hidden scene object therefore reduces. In the far-field limit, the penumbra weakens significantly and the irradiance distribution of the ceiling is nearly constant. Moreover, at extremely large ranges, objects may become undetectable and near-impossible because the useful signal is extremely weak.
- The perceived size of the object relative to the observation plane decreases and becomes increasingly closer to being approximated by a point source. This means the produced penumbra starts to sharpen becoming a sharp shadow in the limit. Sharper shadows make shape reconstruction harder, but localization in angle becomes marginally easier.
- The drop in irradiance due to radial falloff also makes range harder to estimate from the measurements.

In the case where object depths and sizes increase correspondingly to keep their subtended solid angle fixed:

- The objects receive the same amount of light overall, by being larger, yet are less luminous from the perspective of the ceiling plane. The irradiance of a ceiling patch in the measurement FOV decreases, as a consequence, but less dramatically than in the first case (above).
- The sharpness of the penumbra does not change, however, its overall brightness does decrease. This makes shape reconstruction more challenging in noise and situations with visible side ambient illumination.

In sum, the difficulty of the reconstruction problem does vary with the size of the actual scene: nearer objects are easier to reconstruct, as are larger objects generally. Some challenges—such as reduced signal strength caused by penumbra being weaker and the variation of the irradiance distributions being milder for far-field objects—that arise due to increased scene size can be possibly alleviated by a number of mechanisms. For instance, using more intense illumination of the hidden scene, using detectors with greater sensitivity and dynamic range, by increasing the size of the measurement FOV (and commensurately increasing the camera resolution, so that the projected pixel sizes are unchanged). Multiple exposures can help improve SNR in high-noise scenarios, especially when used in combination with the foregoing mechanisms.

Ultimately, based on the achieved performances/reconstructions obtained in ambient illumination shown in Figure 4(d), we speculate that the current setup could be extended to typical room-sized scenes of around $3 \times 3 \times 2 \text{ m}^3$, and human-sized occupants, with little degradation in performance (especially when there is no additional visible side ambient illumination).

We have included an abridged version of the above response in the discussion section of our manuscript.

Comment 1.8

Overall, I enjoyed reading this manuscript, it is very high quality work, it is novel and does represent a significant step forward in the NLOS imaging field. I would therefore recommend publication as is or ideally, with some minor modifications following the suggestions above assuming that all of these make sense.

Response:

Thank you. Your suggestions have helped to further strengthen our work.

2 Response to Reviewer #2

Comment 2.1

Key Results:

This paper by Czajkowski and Murray-Bruce presents the first passive NLOS method to produce compelling 3D results. The method requires two key innovations and some reasonable signal processing assumptions. The first major innovation is the use of doorframe as an occluder that is sufficiently ubiquitous in indoor settings to be useful, and which crucially enables high angular resolution in a second dimension, compared to existing “corner camera” approaches. Secondly, the authors propose a novel coordinate system that yields critical separability in the two angular dimensions. This coordinate system description allows the authors to derive the light propagation model in a simplified way that facilitates more straightforward inversion. Taking advantage of doorframe occluders that exist “in the wild,” the two-edge-resolved NLOS imaging configuration is able to reproduce full color 3D images of macroscopic scenes with consumer-grade equipment and reasonably high accuracy.

Response:

This is an accurate summary. Thank you for taking the time to fully evaluate our work.

Comment 2.2

Validity:

The data acquisition and processing approaches are largely valid. I have some suggestions below that may lead to slightly better results, but the approach here is principled and reasonable.

Response:

We appreciate the suggestions, they inspired improvements in our results.

Comment 2.3

Significance:

These results are a significant step forward for the field. Current assumptions are that high-quality 3D NLOS results require expensive, specialized equipment. This paper demonstrates that there is significant 3D information that can be extracted from only a standard camera.

Response:

Thank you for the highly positive comment on the significance of this work. We are eager to further explore the possibilities of passive NLOS imaging.

Comment 2.4

Clarity and context, References:

The paper is clearly written and properly cites the relevant contributions in the literature.

Response:

We are delighted to receive this feedback.

Comment 2.5

Data and methodology, analytical approach: A detailed discussion of the datasets and methodology are discussed below. Acquisition strategy - Like most NLOS demonstrations, the setting is somewhat contrived, with a few objects of interest within a minimally reflective background and with a doorframe opening. However, using multiple (orthogonal) edges of a doorframe is a clever approach to passive NLOS with some clear advantages of the two-edge-occluder camera. First, the head of the doorframe naturally blocks light sources on the ceiling, which are a major source of interference in other passive settings. Furthermore, although other work with 2D or 3D occluders have shown the best results for passive NLOS imaging, the occluder shape either must be known or estimated. Having an (approximately 2D) occluder that is visible to the observer is both less contrived and also eliminates one of the challenges of existing passive methods.

Response:

We appreciate your balanced summary of the advantages and limitations of our demonstration. Based on additional experiments—that were motivated by this comment and others appearing further along—we believe that deviating from scenes with minimally reflective backgrounds is possible. Some preliminary reconstructions are presented for rooms with all-white walls and floors in our response to Comment 2.11 below. We hope that dissemination of our work will encourage others to explore this limitation more thoroughly, to complement our planned future explorations as well.

Comment 2.6

Scene representation model - It took reading the supplementary material in detail to understand why the projected elevation coordinate system makes such a large difference in the reconstruction. Perhaps more intuition can be given for why this system is a natural fit for TERI and significantly better than spherical coordinates. If one considers a single edge occluder, then cylindrical coordinates are a natural fit, as have been explored in several papers. Using two orthogonal edge occluders effectively yields a pair of single edge cameras with complementary information in orthogonal directions. (Can one think of these coordinates as somehow a compromise between two complementary cylindrical coordinate systems?) The projected elevation system allows one to describe angular coordinates from each of the edge occluders separately, whereas using spherical coordinates forces the elevation description to depend on the azimuth.

Response:

These are great observations. One can absolutely think of the proposed projected-elevation spherical coordinate (PESC) system as a combination of two complementary cylindrical coordinate systems. The combination is such that: the angular coordinates in the two cylindrical systems form the two angular coordinates of PESC, while the range axis of PESC is a function of the two range-height pairs of the cylindrical coordinate systems. Similarly, PESC may also be seen as a combination of two complementary *polar coordinate systems*, where each system pivots about one edge of the doorway.

We have revised the paragraph in the main manuscript that explains the intuition behind the projected-elevation spherical coordinate system. The new addition reads:

To achieve information orthogonality, the proposed coordinate system is constructed such that its angular coordinates, (θ, ψ) , each pivot about one of the vertical and horizontal occluding edges of the doorway. Much like polar (or cylindrical) coordinates are natural for the corner camera³¹ and its variants^{32,36,39}, the projected-elevation spherical coordinate system is natural for TERI. Because single-edge corner cameras exploit a vertical occluding edge, then for a point source in the hidden scene, the most significant changes in the observed penumbra occur when the point source moves in azimuth alone (pivoting about the vertical edge). Range-only motion produces only subtle intensity changes due to radial falloff effects, while the overall penumbra shape is unchanged. The projected-elevation spherical coordinate system, therefore, represents a combination of two orthogonal corner camera systems resulting from two orthogonal edges of the doorframe. In this two-edge configuration, a hidden scene point source will produce a sharp shadow resembling the top and slanted edges of a right

trapezoid (see example observations shown in Figures 1d and 1e). Representing the hidden scene point in the proposed coordinate system, changes in the slope of the slanted edge of the shadow are due exclusively to point source motion along azimuth θ (with pivot at the vertical occluding edge). Conversely, changes in the position of the top shadow edge are exclusively due to point source motion along the projected elevation angle ψ (with pivot at the horizontal occluding edge). Moving along the range axis of the projected-elevation coordinate system does not cause any changes in the penumbra's shape; only small intensity changes occur due to radial falloff. This property ensures that errors in any one of the angular coordinates will not affect the other. Because the elevation is defined in the plane containing the azimuth, this phenomenon does not occur in a spherical coordinate system. Further justifications of this phenomenon, including experimental and theoretical demonstrations, are provided in the supplementary document.

Comment 2.7

- This separation greatly simplifies the forward modeling, which I assume makes the reconstruction problem much easier. There is a hint to the advantage in the mention of the information orthogonality, which seems fascinating and potentially critical to actually being able to solve the inverse problem, but more intuition/explanation should be given to explain how that makes a meaningful difference. (i.e., now that we see the angular coordinates are uncoupled in the new coordinate system, what are the downstream effects for forward model/inverse problem?)

Response:

Nice point again, thanks for looking thoroughly through the supplementary document. The excerpts below now appear in the manuscript.

This apparent decoupling of how changes in θ and ψ impact the measurement ensures that errors in any one of the angular coordinates will not impact our estimation of the other.

⋮

Existence of information orthogonality among the range, azimuth, and project-elevation coordinates is crucial to the success of our two-step reconstruction procedure. Notably, it ensures that accurate angle estimation (i.e., shape) is achievable without knowing the ranges of objects within the hidden scene because errors in one do not prevent reliable reconstruction of the other. Additionally, the proposed coordinate system also simplifies the forward modelling by enabling closed-form expressions for incorporating the effect of occlusion. Consequently, we could accurately compute the forward model matrix without performing computationally intensive numerical integrations.

Comment 2.8

Reconstruction method - The approach of solving for shape first and then range is interesting. I'm surprised that initialization with a single depth is good enough to recover shape reasonably well. Are there any constraints on how the initial depth is selected?

Response:

For all reconstructions in the main manuscript (and most in the supplementary document), we chose the initial range ρ_0 to be either 0.5 m or 1 m arbitrarily. We had no theoretical or practical constraints in selecting the initial range. This, seemingly surprising phenomenon, is a natural consequence of the information orthogonality property of our chosen coordinate system (and secondarily, the extremely mild dependence of the measurement signal on the range of surface elements). In theory, exact information orthogonality would mean that there are no constraints on how the initial range/depth is selected. In practice, however, the non-diagonal terms of the Fisher information matrix (**shown in Supplementary Figure 10**) although extremely small, are not precisely zero. Thus, errors in any coordinate (e.g., range)

only mildly impact the others (azimuth and projected elevation). This comment inspired a compelling and illustrative investigation that we had not previously considered. Shown, in the figures that follow, are shape-only reconstructions (i.e., completing only Step 1 of the TERI algorithm) using various range initialisations between 0.1 m and 10 m. The reconstructions show almost complete invariance with range values across two orders of magnitude. The most significant differences are increased amounts of clutter attempting to explain extraneous ambient illumination due to multiple reflections. (These differences become almost inconsequential when the entire TERI algorithm concludes. We did not attempt to tune the regularization parameter λ in these reconstructions, and the same value was used in almost all reconstructions. Thus, proper tuning of λ may mitigate the differences.) These figures and an abridged version of the above text have been added to Section S5.4 of the supplementary document.

$$N_\theta = N_\psi = 90$$

$$N_\theta = N_\psi = 60$$

$$\rho_0 = 1$$

$$\rho_0 = 2$$

$$\rho_0 = 5$$

$$\rho_0 = 10$$

Invariance of TERI reconstructions to arbitrary range initialization. Shape-only reconstruction results of solving the LIP in Step 1 of the TERI algorithm for different choices of initial range values ($\rho_0 = 0.1, 0.25, 0.5, 0.75, 1, 2, 5, 10$). Two different reconstruction resolutions are also shown: left is for $N_\theta \times N_\psi = 90 \times 90$ and right is for $N_\theta \times N_\psi = 60 \times 60$.

Comment 2.9

- After recovering the shape, estimating the range is the more difficult problem to solve, although there is some noticeable improvement based on improving the range estimate. I wonder whether there would be any benefit in alternating between the shape/range estimation procedures. Could the shape be refined from repeating that step even once after the range has been corrected?

Response:

This is a nice question. We had considered and implemented this previously but observed minor improvements. We feel that this also highlights a key strength of our information-orthogonal projected-elevation spherical coordinate system. Because our three coordinates θ , ψ , and ρ are information-orthogonal, errors in one dimension do not impact the other dimensions. In the first step, we are only reconstructing angular estimates, thus, any errors in ρ have little impact on the reconstruction.

However, because range and radiosity are coupled, radiosity can be slightly improved by an update step. While keeping the ℓ_1 -prior brings minimal improvement as shown in the following reconstructions (top views shown), greater improvement can be obtained by switching to a total variation (TV) prior which enforces spatial smoothness. We, therefore, updated the TERI algorithm to incorporate a TV prior in Step 2. The new algorithm and resulting reconstructions have been incorporated into the revised manuscript. We believe this update has greatly improved the colour rendition and overall visual accuracy of our reconstructions. We hope you will agree. A sample of the reconstructions is shown in our response to Comment 2.13 below.

We do, however, take inspiration from this comment for an algorithm that not only re-computes the shape, but one that also re-computes the intensity after the range reconstruction. We implement an algorithm here that, for each object, pads the angular span reconstructed from Step 1 and applies the determined range from Step 2. We then form another angular reconstruction with each angular facet placed at the estimated range.

Shape and Intensity Refinement The left column shows a topview of 3D reconstructions before shape enhancement (the original algorithm) and the right shows after shape and intensity re-estimation using an ℓ_1 prior. (Axis units are in meters.) There is, overall, only marginal change in shape; however, we do notice a difference in the computed intensities. Notice that the U in the USF appears brighter in the refined reconstruction. This makes some sense because the U is closer to the overhead light, and is therefore actually brighter in the original scene.

Comment 2.10

- The assumption of all points in a cluster being located at the same depth irks me a bit. I understand the authors have tried to justify this by showing that estimating a range for every point in the scene would have far to little information for robust estimation. Figure S8 in the supplement justifies that the CRB is higher for more clusters. However, the CRB is a lower bound on the variance of an *unbiased* estimator. I would argue that assuming all points in a cluster are located at a fixed range is a biased estimator, and subsequently breaking a cluster into progressively smaller subclusters allows for lower bias but higher variance. With no clusters at all (independent range estimate per pixel), the mean-squared error is understandably far too high. However, it's not obvious that for every scene the high-bias, low-variance estimate of a single cluster will yield lower MSE than the lower-bias higher-variance estimate of a small number of clusters. This is especially true for extended objects such as the shelf, which is visibly distorted in Fig. 3b. I wonder whether a slightly more sophisticated method would decide on clusters based not only on connected components, but also how compact those components are in 3D. So after the 2 steps (shape/range estimation) in the existing algorithm, one could check whether the single-cluster assumption seems reasonable. Using the single-cluster estimate as the initialization, smaller subclusters could then refine the 3D position estimate for more realistic results.

Response:

We agree entirely with your comments. Our clustering algorithm could be improved by subsequently splitting large clusters into smaller subclusters, we had briefly mentioned this in our original submission. We have updated the discussion section of the manuscript to elaborate further. Furthermore, we include two new preliminary experiments (using real measurements) and further discussions of a more sophisticated clustering algorithm in the supplementary document. The new algorithm successively divides a large cluster into smaller sub-clusters and its effectiveness is demonstrated in two scenarios:

- **Single-object scenario (Figure 1).** This scenario explores the reconstruction of a single large planar piece of card. The aim is to show that for an object that occupies a single range, segmenting it into smaller sub-clusters has little impact on the range estimate.
- **Two-object scenario (Figure 2).** This scenario explores the reconstruction of two planar objects that are separated in range but appear as neighbours in azimuth and projected elevation angles. The aim is to show that subdividing the large cluster yields more accurate (reduced bias) range reconstructions.

We provide the reconstructions here, in Figures 1 and 2 below, and leave the full details of the experiment and algorithm in Supplementary Section 7. For the single-object case, whose final reconstruction (with the TV-based radiosity refinement) is shown in Figure 1(e), it can be seen that the ranges of the sub-clusters are relatively close to each other, as well as the ground truth value when estimating their ranges independently. The algorithm correctly returns the original cluster instead of smaller sub-clusters.

For the two-objects case, the final reconstruction, shown in Figure 2(e), highlights the effectiveness of the preliminary cluster refinement scheme. The new ranges are highly accurate with significantly reduced bias.

Implementing a more sophisticated clustering approach algorithm that decides whether to recursively split a cluster based on its compactness is a compelling idea. Ultimately, we believe judicious clustering would require further thorough investigations and is an interesting future line of research.

Side view

(a) Photograph of the hidden scene.

(b) Recon. after Step I of TERI.

(c) Recon. after Step II of simplified TERI algorithm (estimated range $\hat{\rho} = 1.00$ m).

(d) Recon. after line 2 of the prelim. cluster refinement (sub-cluster ranges 1.064, 1.099, 1.150, 1.161 m).

(e) Final recon. after convergence of the prelim. cluster refinement, and TV radiosity refinement.

Figure 1: **Scene reconstruction with cluster refinement algorithm for single-object scene.** (a) The scene is a rectangular card whose measured range is 1.04 m. (b) The estimated shape of the hidden scene after Step I of TERI concludes. (c) The initial reconstruction of the hidden scene after Step II of the simplified (no TV) TERI algorithm; the estimated cluster range is 1.00 m. (d) Intermediate reconstruction after splitting the cluster and updating their range estimates (i.e., after running lines 1 and 2 of the *preliminary cluster refinement algorithm*); estimated sub-cluster ranges are 1.064, 1.099, 1.150, 1.161 m. (e) Final reconstruction after convergence of the *preliminary cluster refinement algorithm*, followed by TV-regularized scene radiosity update.

View from origin

Side view

(a) Photographs showing different views of ground truth hidden scene.

(b) Recon. after Step 1 of TERI.

(c) Recon. after Step II of simplified TERI algorithm (estimated range $\hat{\rho} = 0.73$ m).

(d) Recon. after line 2 of the prelim. cluster refinement (est. sub-cluster ranges 1.24, 1.32, 1.67, 1.64 m).

(e) Final cluster recon. after convergence and TV radiosity refinement (est. ranges 0.99 m and 1.48 m).

Figure 2: **Scene reconstruction with cluster refinement algorithm for a two-object scene.** (a) The scene is two rectangular cards whose measured ranges are 1.04 m and 1.55 m. (b) The estimated shape of the hidden scene after Step I of TERI concludes. (c) The initial reconstruction of the hidden scene after Step II of the simplified (no TV) TERI algorithm; the estimated cluster range is 0.73 m. (d) Intermediate reconstruction after splitting the cluster and updating their range estimates (i.e., after running lines 1 and 2 of the *preliminary cluster refinement algorithm*); estimated sub-cluster ranges are 1.24, 1.32, 1.67, 1.64 m. (e) Final reconstruction after convergence of the *preliminary cluster refinement algorithm*, followed by TV-regularized scene radiosity update.

Comment 2.11

- The tested scenes, with a small number of colorful objects surrounded by a black background, encourage the use of the sparsity prior (i.e., ℓ_1 -norm regularization) for reconstruction. What are the limitations of this assumption? Would it be possible to reconstruct an empty room with white walls, which would reflect light from all angles? What about a white object in front of a white wall, or more generally any object partially occluding an object of the same color?

Response:

In the original draft, our mindset was that our core framework had the flexibility of being compatible with a broad selection of hand-crafted regularizers, and possibly learned priors too. Hence, we devoted little attention to discussing the strengths and limitations of our chosen prior. This flexibility enabled the recent version TERI that uses an ℓ_1 regularizer in the first step (shape reconstruction) and a TV regularizer in the second (to encourage smoothness in radiosity estimates).

We now include text that draws out these limitations in the discussion section of the main manuscript, as well as real experiments for two new scenes in Supplementary Section 5.5. The new experiments, whose results are also shown here in Figures 3 and 4, demonstrate the ability of our algorithm to correctly all solid angles containing reflective surface elements (even when scenes are not sparse), and surprisingly reveal hidden scene shadows (Figure 4). Despite this, it is challenging to intuit the scene configuration from the 3D visualisations, especially when a foreground object exists and takes the same colour as the background walls.

(a) Photographs showing different views of ground truth hidden scene.

(b) Different views of the reconstructed scene.

Figure 3: **Example reconstruction of a non-sparse scene.** (a) The hidden scene is an empty room with white walls and floor. The wall parallel to the doorway is placed at 0.67 m from the doorway plane, the other wall is at 0.60 m from the xz -plane, and the floor is 1.24 m from the doorway head. (b) 3D view, side, and top views of the computed reconstruction. The entire scene is reconstructed as a single large cluster of surface elements with an estimated range of 0.79 m. One of the walls is reconstructed as being brighter than the other because the hidden scene illumination was facing that wall.

(a) Photographs showing different views of ground truth hidden scene.

(b) Photographs showing different views of ground truth hidden scene.

Figure 4: **A non-sparse scene with partial occlusion.** (a) The hidden scene configuration, which shows a room with a white and blue broom inside it. From the perspective of the ceiling plane, the broom casts a shadow on some portions and fully occludes some other portions of the room’s walls and floor. The wall parallel to the doorway is placed at 0.67 m from the doorway plane, the other wall is 0.60 m from the xz -plane, the floor is 1.24 m from the doorway head, while the broom is roughly 0.53 m from the origin. (b) The obtained reconstruction of the scene shows a 3D view along with side and top views. The entire scene is reconstructed as a single large cluster of surface elements with an estimated range of 0.76 m. The blueness and orientation of the broom’s bristles and part of the shadow cast by the broom are visible in the reconstructions.

Comment 2.12

- I don’t quite understand the background light modelling. For the near-field contribution from a point source on the visible side, is it assumed the point source location is known? Or does b_{vis} allow for a point source at any location (which might collapse the linear independence with the all-ones vector)?

Response:

The true locations of any light sources on the visible side are not assumed to be known. Rather, we assume that these visible side contributions can be approximated by a linear mixture of an all-ones vector and the illumination pattern created by a point source in the near-field. The point source’s location is arbitrarily chosen to lie on the visible side. With the all-ones vector, we aim to account for approximately uniform contributions to all pixels of the measurement FOV, and with the point source we aim to account for contributions with slow and non-penumbal variation over the measurement FOV. In all experiments, we used $(x, y, z) = (-0.4, -0.4, 2)$.

Moreover, experimentally, we observed that in situations with ambient illumination, hidden scene surface elements near the doorway dominate the reconstructions, and appear extremely bright because they are attempting to explain background contributions that affect all pixels. When the background

columns are added, the obtained reconstructions use them to explain the background contributions over using the hidden scene surface elements.

Comment 2.13

- For the reconstruction display, the colors seem skewed relative to the true photographs. In particular, yellow objects appear to be far more red than the true objects. I think the problem is that the raw RGB values are being mapped directly to the 3 color channels of the point cloud display. I wonder whether first multiplying by a color conversion matrix (see doi:10.1117/1.OE.59.11.110801) and/or white balancing would yield more perceptually consistent results.

Response:

First and foremost we would like to thank the reviewer for this comment, which was a concern also raised by Reviewer 1. We took this comment to heart and investigated many possible remedies. White balancing and utilizing a colour conversion matrix were great suggestions, which we attempted carefully. Yet, we had little success with both approaches, even when we recollected additional experimental datasets in more controlled conditions. What brought the greatest improvements, however, was utilizing a total variation (TV) prior in place of the ℓ_1 prior. Below, we show the old and new reconstructions side-by-side, for two example scenes. Accordingly, the manuscript has been updated with the improved TERI algorithm and resulting reconstructions.

Comparison between TV-regularized update vs No update (original algorithm). Notice the blue portions of the doughnought are better reconstructed. The beige-coloured arms and legs of the red-green T-pose mannequin are clearly visible and distinguishable from the red and green parts of the mannequin.

Comment 2.14

- How are the threshold gamma and number of clusters J_{max} chosen?

Response:

Generally, the effects of the tuning parameters γ and J_{\max} on the reconstructions are mild. In all reconstructed scenes, $\gamma = 0$ was used, while J_{\max} was set to either 4 or 5.

The higher γ is, the more fragmented the reconstruction will be, which creates more clusters. In tuning γ , the aim is to strike a balance between: (i) ensuring that two separated objects do not become linked in the reconstruction by a spurious surface element due to clutter (i.e., γ too small); and, (ii) ensuring that an object is not incorrectly split into many multiple clusters (i.e., γ too large). J_{\max} should be chosen to be larger than the number of anticipated objects in the ground truth scene. If J_{\max} is much higher than the true number of objects, in a low noise setting it becomes an inactive constraint and has no effect; however, in a noisy scenario with significant background clutter then this could allow for many individual noise surface elements to be counted as objects, which generally still does not impact reconstruction quality, but does increase the runtime.

Comment 2.15

Additional Comments - I find Fig. 1b to be hard to follow. I understand the idea of the projected elevation, but something about the 3D structure projected onto 2D of the page is hard to interpret. Perhaps using a wireframe cube to show the 3d position in the Cartesian coordinates would help make the projection of the point onto the xz plane easier to interpret.

Response:

Thanks for the wireframe cube suggestion. We adopted the idea and believe it has helped to improve the clarity of Figure 1b (as well as Figures 1d and 1e).

Comment 2.16

- How is the projected elevation coordinate related to the classical elevation coordinate?

Response:

Mathematically, $\tan(\psi) = \cot(\varphi)\sec(\theta)$, based on the convention for elevation ϕ and projected elevation ψ indicated in main manuscript Figure 1. This relationship is now explicitly stated in Figure 1 of the main text.

Comment 2.17

- Lines 38-41: Not clear whether this is still referring to ref. 27.

Response:

The wording has been fixed to make it more clear. Thanks.

Comment 2.18

- Figure 2 caption: "Four such clusters were identified, one for each object" → "one for each of the three objects"

Response:

Changed!

Comment 2.19

- Line 316: In what way is the chair + basketball an exception to the modeling?

Response:

The facets of the projected elevation spherical coordinate system are pointing toward the origin. Because the backrest of the chair does not point toward the origin, and because the curvature of the basketball is not toward the origin, both objects deviate slightly from the model. Nonetheless, this mismatch can be lessened with finer angular discretizations. We have adjusted the wording in the manuscript to make this more clear.

Comment 2.20

- Several of the references on arxiv have appeared now in journals/conferences

Response:

Thanks. The references have been updated.

Comment 2.21

- Supplement Fig. 3: what does the coloring of the surface element represent?

Response:

The colors in Supplementary Fig. 3 hold no technical significance. We found that the color gradient helped with interpreting the intended 3D visualization. A comment has been to the captions of Supplementary Figs. 2 and 3 to clarify this.

Comment 2.22

- Supp Fig. 6:
 - o subfigures not labelled
 - o The improvement in CRB with more edges is remarkable! Very illustrative result.

Response:

The subfigures have been labeled. Thanks also for the comment.

Comment 2.23

- Perhaps I missed it, but what is the d in Supp. Eqn. (41)?

Response:

Thank you for catching this error, d is now explicitly defined in the Supplementary Document. It refers to the real-world length of a square pixel in the observation photograph.

Comment 2.24

- Supp. Fig. 14: Some colored text markup should be removed.

Response:

Done.

REVIEWERS' COMMENTS

Reviewer #1 (Remarks to the Author):

The authors have replied to all my questions very carefully. I had already provided a positive assessment of this very good work. This is very high quality research that represents a significant and very interesting step forward in non line of sight imaging. I recommend acceptance without any need need for further revision.

Reviewer #2 (Remarks to the Author):

I thank the authors for their extremely thorough response to the previous reviews. They have gone above and beyond to address each point with additional experiments, new processing, and clear analysis. I particularly appreciate the authors' willingness to describe limitations and avenues that were tested but unsuccessful (e.g., color correction). I'm pleased to see that the TV regularization was able to markedly improve the color consistency. I also found the investigations of information orthogonality, non-sparse scenes, and sub-cluster splitting to be compelling.

Given my previous positive comments about the significance of the work, as well as the revision completely addressing my concerns, I can now fully support publication of the manuscript.